# Ultrafast decoupling of polarization and strain in ferroelectric BaTiO₃

Le Phuong Hoang[1,2,3], David Pesquera [4], Gerard N. Hinsley [5], Robert Carley [1], Laurent Mercadier [1], Martin Teichmann [1], Elena Martina Unterleutner [6], Daniel Knez[6], Martina Dienstleder[7], Saptam Ganguly [4], Teguh Citra Asmara [1], Giacomo Merzoni[1,8], Sergii Parchenko [1], Justine Schlappa [1], Zhong Yin [1,13], José Manuel Caicedo Roque[4], José Santiso [4], Irena Spasojevic [9], Cammille Carinan[1], Tien-Lin Lee [10], Kai Rossnagel [3,11], Jörg Zegenhagen [10], Gustau Catalan [4,12], Ivan A. Vartanyants [5], Andreas Scherz [1] & Giuseppe Mercurio [1] ✉

A fundamental understanding of the interplay between lattice structure, polarization and electrons is pivotal to the optical control of ferroelectrics. The interaction between light and matter enables the remote and wireless control of the ferroelectric polarization on the picosecond timescale, while inducing strain, i.e., lattice deformation. At equilibrium, the ferroelectric polarization is proportional to the strain, and is typically assumed to be so also out of equilibrium. Decoupling the polarization from the strain would remove the constraint of sample design and provide an effective knob to manipulate the polarization by light. Here, upon above-bandgap laser excitation of the prototypical ferroelectric BaTiO₃, we induce and measure an ultrafast decoupling between polarization and strain that begins within 350 fs, by softening Ti-O bonds via charge transfer, and lasts for several tens of picoseconds. We show that the ferroelectric polarization out of equilibrium is mainly determined by photoexcited electrons, instead of the strain.

Ferroelectric materials are characterized by many properties, including piezoelectricity and pyroelectricity, besides ferroelectricity, which make them attractive for a wide range of applications, such as non-volatile memories, transistors, sensors, and actuators[1,2]. The key property of a ferroelectric material is the ability to switch its spontaneous polarization in response to an external electric field. This is typically achieved by a static or pulsed electric field with the consequent limitations given by the need for complex circuitry and switching times of hundreds of picoseconds to nanoseconds[3]. These

challenges can be overcome by optical control of the ferroelectric polarization. Light-matter interaction enables remote and wireless control of the ferroelectric polarization on the picosecond timescale[3]. Moreover, since all ferroelectrics are also piezoelectrics, the ferroelectric polarization is strongly coupled to the strain, i.e., the lattice deformation[4]. Optical control of polarization and strain has been achieved in several cases. For example, in multilayers of ferroelectric and electrode thin films, an optical laser was used to excite the metal (or semiconductor) layer and indirectly the ferroelectric material,

[1]European XFEL, Schenefeld, Germany. [2]Max Planck Institute for the Structure and Dynamics of Matter, Hamburg, Germany. [3]Institute of Experimental and Applied Physics, Kiel University, Kiel, Germany. [4]Catalan Institute of Nanoscience and Nanotechnology (ICN2), CSIC and BIST, Campus UAB, Bellaterra, Spain. [5]Photon Science, Deutsches Elektronen-Synchrotron DESY, Hamburg, Germany. [6]Institute of Electron Microscopy and Nanoanalysis (FELMI), Graz University of Technology, Graz, Austria. [7]Graz Centre for Electron Microscopy (ZFE), Graz, Austria. [8]Dipartimento di Fisica, Politecnico di Milano, Milano, Italy. [9]Department de Física, Universitat Autònoma de Barcelona, Bellaterra, Spain. [10]Diamond Light Source Ltd., Didcot, Oxfordshire, UK. [11]Ruprecht Haensel Laboratory, Deutsches Elektronen-Synchrotron DESY, Hamburg, Germany. [12]Institut Català de Recerca i Estudis Avançats (ICREA), Barcelona, Spain. [13]Present address: International Center for Synchrotron Radiation Innovation Smart (SRIS), Tohoku University, Sendai, Japan. ✉e-mail: giuseppe.mercurio@xfel.eu

leading to a transient modification of the strain[5–8], or the polarization by charge redistribution at the interface[9]. In other studies, light was absorbed directly by the ferroelectric material, inducing changes in the spontaneous polarization[10,11] or lattice strain in clamped[12–18] or freestanding[19] ferroelectric thin films. THz light was employed to rotate[20] or even transiently reverse the orientation of the spontaneous polarization[21]. In all these studies so far, either the polarization or the strain was measured, and a direct proportionality between sponta- neous polarization and strain was typically assumed[22]. This pro- portionality is based on the piezoelectric effect, which is well captured by the Landau-Ginzburg-Devonshire theory when the lattice distortion is along the polarization axis[23]. While this assumption is valid under equilibrium conditions, as demonstrated experimentally, e.g., in refs. 24,25, it may not hold under out-of-equilibrium conditions following light-matter interaction. Decoupling the polarization from the strain would remove the constraint of sample design[24,26] or strain tuning[27,28] to achieve specific properties, and, at the same time, would provide a more effective and ultrafast knob to manipulate the polarization by light.

　　To explore this scenario, we probe the out-of-plane strain and the spontaneous polarization of the prototypical ferroelectric BaTiO$_3$ upon above-bandgap absorption of ultrashort UV light pulses with a peak power intensity of a few tens of GW cm$^{-2}$ (Fig. 1a). A fundamental understanding of the relationship between strain and ferroelectric polarization out of equilibrium requires their investigation on their natural timescale encompassing $\approx$ 100 fs to several tens of ps. We employ a combination of time-resolved X-ray diffraction (tr-XRD), time-resolved optical second harmonic generation (tr-SHG), and time- resolved optical reflectivity (tr-refl) to obtain the magnitudes and the separate dynamics of the out-of-plane lattice parameter, the

spontaneous polarization, and the photoexcited carrier density, respectively[29], with a time resolution of $\approx$ 90 fs. In this paper, we show the mechanisms that govern the structure and polarization changes in a ferroelectric material and their complex relationship out of equili- brium in the presence of photoexcited electrons and lattice defor- mation. In particular, since the strain wave propagates at the speed of sound, whereas electronic interactions are much faster, we induce and measure an ultrafast decoupling between polarization and strain, which we assign to the photoexcited electrons. First, we present the lattice response to the absorption of UV laser pulses and the corre- sponding data modeling. Next, we present tr-SHG and tr-refl data. Finally, we bring together all the results and discuss the underlying physical mechanisms in the context of hitherto known phenomena taking place in ferroelectric materials.

## Results
### Photoinduced structural dynamics
Our sample consists of a coherently strained, monodomain BaTiO$_3$ (BTO) thin film, grown on a GdScO$_3$ (GSO) substrate, with a SrRuO$_3$ (SRO) bottom electrode sandwiched in between (see "Methods"). Under a compressive strain of $-0.55\%$ imposed by the substrate, the BTO film shows an out-of-plane ferroelectric polarization $\mathbf{P}_s = P_s\mathbf{z}$ pointing toward the sample surface (Fig. 1a), where $P_s$ is the magnitude of the polarization and $\mathbf{z}$ is the unit vector along the out-of-plane direction. The sample is excited above the BTO band gap $E_g = 3.4$ eV[30] using 266 nm laser pulses, with photon energy $E = 4.66$ eV, at an inci- dent pump laser fluence of 2.7 mJ cm$^{-2}$. Time-resolved X-ray diffraction of the (001) Bragg reflection is employed to probe the lattice response of our ferroelectric thin film along the out-of-plane direction. The lattice deformations along the in-plane directions on the picosecond

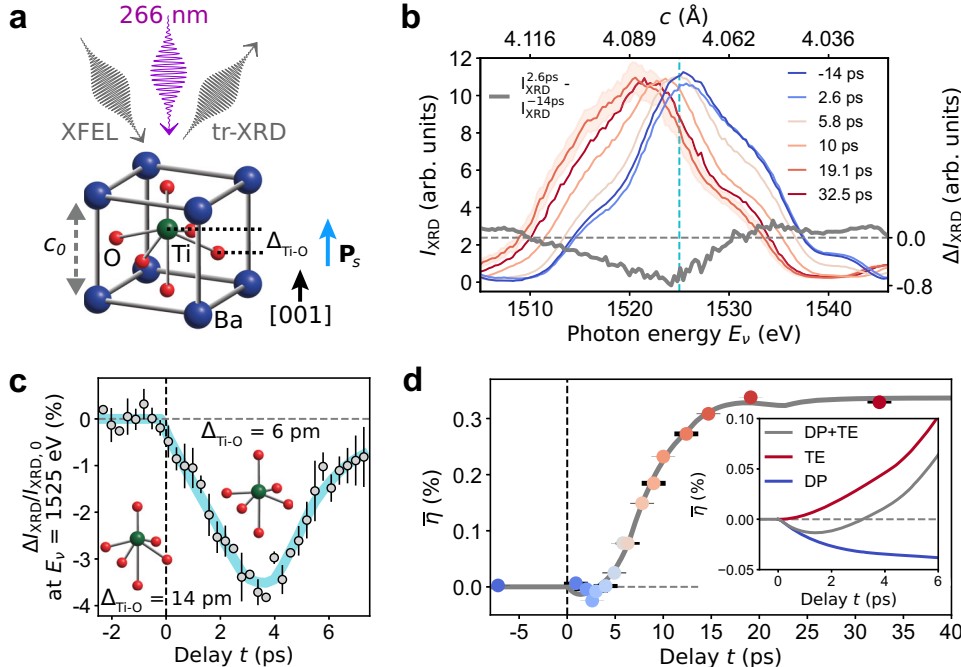

**Fig. 1 | Photoinduced structural dynamics and time evolution of strain.**
**a** Sketch of the BaTiO$_3$ unit cell, the out-of-plane lattice parameter in the ground state $c_0$, the relative displacement $\Delta_{Ti-O}$ of the Ti atom from the center of the O octahedron, the spontaneous ferroelectric polarization $\mathbf{P}_s$, the pump laser at 266 nm, the incident and diffracted XFEL beams. **b** Selected experimental $I_{XRD}(E_\nu)$ curves at different time delays $t$, and the difference $\Delta I_{XRD} = I_{XRD}^{2.6\,ps} - I_{XRD}^{-14\,ps}$ (gray line). The vertical dashed cyan line marks $E_\nu = 1525$ eV. The shaded red area indi- cates the standard deviation value SD $\approx 0.1 \times I_{XRD}$, with similar values also for the diffraction curves at different delays $t$. For clarity, only the error bar of $I_{XRD}^{19.1\,ps}$ is shown. **c** Delay dependence of $\Delta I_{XRD}/I_{XRD,0} = [I_{XRD}(t) - I_{XRD,0}]/I_{XRD,0}$, i.e., the relative

change of $I_{XRD}(t)$ at the photon energy $E_\nu = 1525$ eV with respect to the equilibrium value at negative delays $I_{XRD,0}$. The solid cyan line is a guide to the eye. The error bars indicate the standard deviation of the experimental data in a bin size of 300 fs. Inset: sketch of Ti and O octahedron at negative delays ($\Delta_{Ti-O} = 14$ pm), and at 3.5 ps ($\Delta_{Ti-O} = 6$ pm). **d** Average BTO out-of-plane strain $\bar{\eta}(t)$ as a function of pump-probe delay $t$. The error bars follow from the determination of $c$ from $I_{XRD}(E_\nu) \pm$ SD. The solid gray line is a fit to the data (Supplementary Note 2). Inset: Modeled con- tributions to the average strain $\bar{\eta}(t)$ due to deformation potential (DP) and ther- moelastic (TE) stress (Supplementary Fig. 6).

timescale are negligible, given the large ratio between photoexcited area ($330 \times 240$ μm$^2$) and BTO film thickness $d_{BTO} = 34.5$ nm.

We observe an initial reduction of the tetragonal distortion, which goes hand in hand with lattice compression, then followed by lattice expansion. In particular, Fig. 1b shows the (001) diffraction intensity $I_{XRD}$ of BTO as a function of the photon energy $E_v$ and at different pump-probe delays $t$ from $-14$ ps to 32.5 ps. At $t = 2.6$ ps we observe the following changes to the Bragg peak as compared to the ground state (at $t = -14$ ps): a decrease in the diffraction intensity $I_{XRD}$ near the peak center, and a shift to higher photon energy, which implies a decrease in the out-of-plane lattice parameter $c$, i.e., lattice compression (see gray curve in Fig. 1b). To further explore this initial structural dynamics, we measure the delay dependence of $\Delta I_{XRD}/I_{XRD,0} = [I_{XRD}(t) - I_{XRD,0}]/I_{XRD,0}$, which quantifies the relative change of $I_{XRD}(t)$ at the photon energy $E_v = 1525$ eV of the BTO peak with respect to the equilibrium value $I_{XRD,0}$ at negative delays. We observe a maximum diffraction intensity drop of $\approx 4$% at $t = 3.5$ ps, with up to 99% recovery to the equilibrium value at $t \approx 7$ ps (Fig. 1c and Supplementary Fig. 1). We assign the initial 4% drop and recovery in diffraction intensity to the displacements of atoms within the BTO unit cell (inset of Fig. 1c). Specifically, simulations based on the dynamical theory of diffraction[31] (Supplementary Fig. 2) exclude the Debye-Waller effect and show that a decrease in the displacement $\Delta_{Ti-O}$ between the Ti atom and the center of the O octahedron by 8 pm can model the measured maximum change in peak diffraction intensity.

We focus next on the BTO (001) Bragg peak measured at longer time delays (Fig. 1b). We observe that $I_{XRD}(E_v)$ at $t > 3$ ps are shifted toward lower photon energies, i.e., larger out-of-plane lattice parameters $c$, with respect to $I_{XRD}(E_v)$ at smaller delays $t$. This can be clearly seen from the plot of the BTO out-of-plane strain $\bar{\eta}(t)$, averaged over $d_{BTO}$ (Fig. 1d). Here, $\bar{\eta}(t) = [c(t) - c_0]/c_0$, with $c(t)$ and $c_0$ representing the average $\bar{c}$ at a given $t > 0$ ps and $t \le 0$ ps, respectively (see "Methods"). In Fig. 1d, we find that: (i) the maximum compression of $-0.024$% occurs at $t = 2.6$ ps, (ii) $\bar{\eta}(t)$ increases linearly at a rate of 0.04 %/ps in the range 4 ps $< t <$ 10 ps, and (iii) settles at 0.34% at $\approx 20$ ps.

The model fitting $\bar{\eta}(t)$ data in Fig. 1d is presented in the following. When a photon with energy $E > E_g$ is absorbed in BTO, electrons are photoexcited from the O 2p-derived valence band to the Ti 3d-derived conduction band[32–34]. The thermalization of photoexcited electrons leads to an increase in the electron temperature ($T_e$), and to changes in the electronic system that can be modeled by the variation of the bandgap as a function of the electronic pressure ($\partial E_g/\partial p$)[4]. In turn, a modified electron system affects the interatomic potential, resulting in atomic motions and contributing to the deformation potential stress $\sigma_{DP}(T_e, \partial E_g/\partial p)$. Subsequently, photoexcited electrons transfer part of their excess energy ($E - E_g$) to the phonon system via electron-phonon scattering, increasing the phonon temperature ($T_p$) on the picosecond timescale. This, in turn, induces a lattice expansion dependent on the BTO thermal expansion coefficient ($\beta$), and contributes to the thermoelastic stress $\sigma_{TE}(T_p, \beta)$. The total stress[4,35] $\sigma = \rho v^2 \eta + \sigma_{DP} + \sigma_{TE}$ generates a strain wave $\eta(z, t)$ that propagates through the material of mass density $\rho$ at the longitudinal speed of sound $v$. Given the incident peak power intensity of 39 GW cm$^{-2}$ and other known sample parameters (Supplementary Table 1), we solve analytically the two-temperature model (2TM, Supplementary Note 1) and the lattice strain wave equation (Supplementary Note 2) to obtain the strain $\eta(z, t)$. The 2TM describes electron and phonon temperatures, $T_e(z, t)$ and $T_p(z, t)$, upon absorption of a laser pulse in our sample, thereby accounting for thermal effects (Supplementary Fig. 3). Finally, we calculate the strain $\bar{\eta}(t)$, averaged over $d_{BTO}$, and obtain an accurate fit of the experimental data in Fig. 1d. A similarly good fit of $\bar{\eta}(t)$ data is obtained for incident pump fluence of 1.4 mJ cm$^{-2}$ (Supplementary Fig. 4). The main outcome of our fit model is a negative $\partial E_g/\partial p$ of the order of $\approx 10^{-3}$ eV GPa$^{-1}$ (Supplementary Table 2), in agreement with first-principles calculations[36], with a resulting bandgap decrease of

about 3.2 meV (Supplementary Note 3). The negative $\partial E_g/\partial p$ causes lattice compression along the out-of-plane direction within the first $\approx 3$ ps, when the negative $\sigma_{DP}$ dominates over $\sigma_{TE}$ (inset of Fig. 1d). Conversely, at larger time delays ($t > 3$ ps), $T_p$ increases (Supplementary Fig. 5) and the thermoelastic term becomes the dominant one, leading to an increase of the average out-of-plane strain $\bar{\eta}(t)$ (Fig. 1d and Supplementary Fig. 6). The calculations of lattice temperature, out-of-plane strain and diffraction curves as a function of delay $t$ and distance $z$ from the BTO surface are reported in Supplementary Note 4. The validity of the model employed to fit the strain $\bar{\eta}(z, t)$ data is further corroborated by the good agreement between the experimental BTO (001) diffraction peaks measured at different pump-probe delays $t$ and the corresponding calculated diffraction curves, based on the strain profiles as a function of delay $t$ and distance $z$ from the BTO surface (Supplementary Fig. 7 and Supplementary Fig. 8).

## Photoinduced ferroelectric polarization and electron dynamics

We turn now to investigating the dynamics of the ferroelectric polarization magnitude $P_s$ and of the photoexcited carriers[37–41], upon excitation of the BTO film by the same 266 nm pump laser with fluence 2.7 mJ cm$^{-2}$. Therefore, we perform tr-SHG experiments[29,42] and simultaneously tr-refl in reflection geometry (Fig. 2a). From SHG polarimetry, i.e., the dependence of SHG intensity $I_{SHG}(\varphi) \propto |\chi_{ijk}^{(2)} E(\omega)^2|^2$ on the polarization angle of the probe beam $\varphi$, we learn about the optical tensor elements $\chi_{ijk}^{(2)}$ of a material, and thus its symmetry[42]. By selecting either horizontal ($p$) or vertical ($s$) polarization of the SHG beam, we measure $I_{SHG}^p(\varphi)$ and $I_{SHG}^s(\varphi)$, shown in Fig. 2b, c (blue points) together with the respective fit curves (see "Methods"), which are based on the $4mm$ point group symmetry with the following nonzero tensor elements: $\chi_{zxx}^{(2)}$, $\chi_{xxz}^{(2)}$, and $\chi_{zzz}^{(2)}$.

Upon laser excitation, the $4mm$ symmetry is preserved (orange points in Fig. 2b, c) and the three tensor elements show similar dynamics (Fig. 2d), characterized by a fast fall time with the maximum drop after $\approx 500$ fs and two exponential recovery time constants of $\approx 5.5$ ps and $\approx 45$ ps (Supplementary Fig. 9 for $-1$ ps $< t <$ 30 ps). Interestingly, the tensor elements representative of the induced electric dipole along the out-of-plane direction $z$ ($\chi_{zxx}^{(2)}$ and $\chi_{zzz}^{(2)}$) show a nearly identical time dependence and a larger relative change than $\chi_{xxz}^{(2)}$, which refers to the in-plane induced electric dipole along the direction $x$. The difference between $\chi_{zxx}^{(2)}$ (or $\chi_{zzz}^{(2)}$) and $\chi_{xxz}^{(2)}$ reaches 0.5% after $\approx 500$ fs and decreases in a few tens of picoseconds (Supplementary Fig. 9). A purely thermal effect[43] would cause a uniform change of all tensor elements $\chi_{ijk}^{(2)}$, whereas the measured different dynamics of $\chi_{ijk}^{(2)}$ indicates a time-dependent lattice distortion and/or change in the electronic distribution of non-thermal origin. In fact, TD-DFT calculations[32,33] show that upon charge transfer, the Ti-O$_\parallel$ bonds between Ti and apical O$_\parallel$ atoms (parallel to $P_s$) are weakened more than Ti-O$_\perp$ bonds between Ti and basal O$_\perp$ atoms (perpendicular to $P_s$), as sketched in the inset of Fig. 2d. Consequently, it is intuitive to expect a larger amplitude of the induced electric dipole along the Ti-O$_\parallel$ direction ($z$) with respect to the Ti-O$_\perp$ direction (in the $xy$ plane), as experimentally demonstrated by our data.

The proportionality $I_{SHG} \propto |\chi_{ijk}^{(2)}|^2 \propto |P_s|^2$ gives direct access to the magnitude of the spontaneous polarization[44]. To this end, we measure the relative change $\Delta I_{SHG}^p/I_{SHG,0}^p$ as a function of pump-probe delay $t$ (Fig. 2e), with the polarization of the 800 nm probe beam fixed to the maximum of $I_{SHG}^p(\varphi)$ at $\varphi = 0°$ (Fig. 2b). Simultaneously, we measure the relative change in reflectivity $\Delta R/R_0$ as a function of pump-probe delay $t$ (Fig. 2f). The data in Fig. 2e [f] are well reproduced by a fit function consisting of the sum of three exponential decay terms, with fall [rise] time $\tau_0$, and recovery times $\tau_1$ and $\tau_2$, convoluted with a Gaussian function representing the experimental temporal resolution (Supplementary Note 5). The initial drop in SHG intensity by 10% within 350 fs is followed by $\tau_1^{SHG} = 7.2 \pm 0.5$ ps and $\tau_2^{SHG} = 200 \pm 140$ ps recovery times, resulting in a 2.3% drop at 40 ps (Fig. 2e). At the same

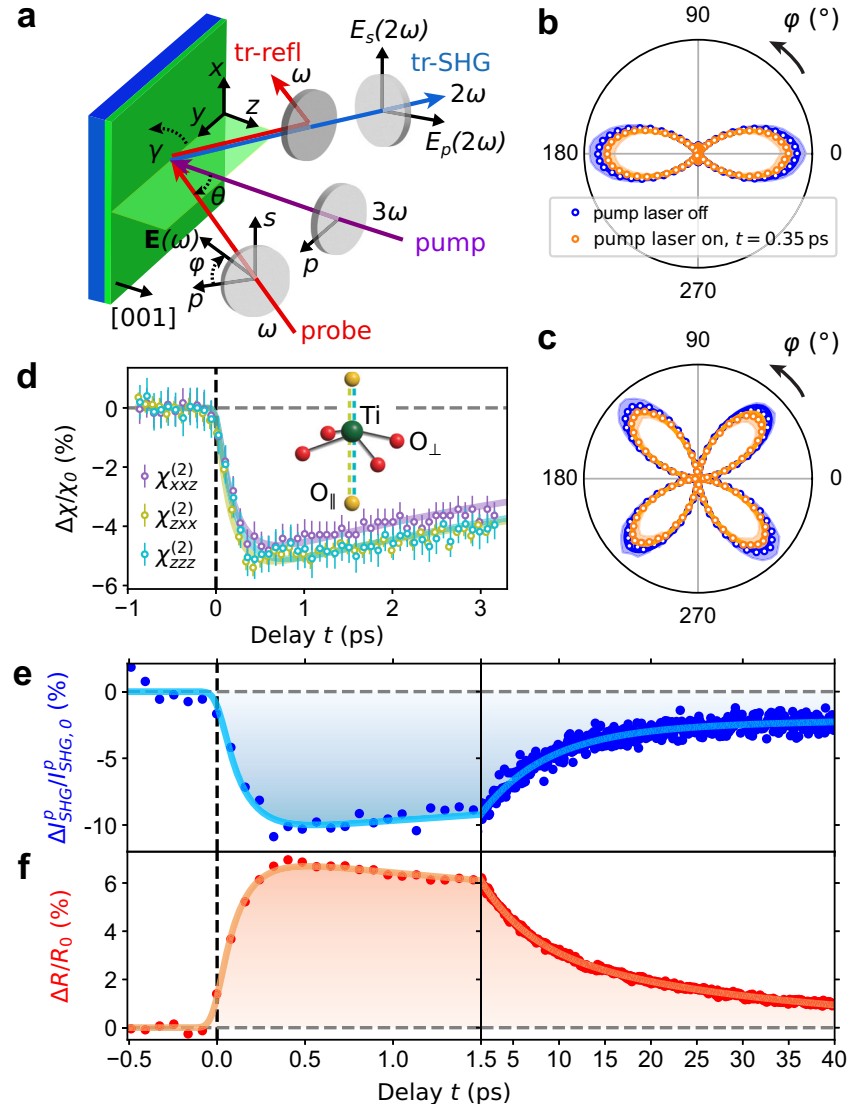

**Fig. 2 | Photoinduced ferroelectric polarization and electron dynamics.**
**a** Sketch of the tr-SHG and tr-refl setup (see "Methods"). **b, c** Polar plots of $I^p_{SHG}(\varphi)$ and $I^p_{SHG}(\varphi)$ measured without pump laser (blue points) and with pump laser at the delay $t = 0.35$ ps (orange points). The orange and blue solid lines are fit curves to the data resulting from equations (1) and (2). The shaded orange and blue areas refer to the standard deviation of the data and amount to ≈ 13%. **d** Relative change $\Delta\chi/\chi_0$ of the tensor elements $\chi^{(2)}_{xxz}$, $\chi^{(2)}_{zxx}$, and $\chi^{(2)}_{zzz}$ as a function of delay $t$, and respective fit curves (Supplementary Note 5). $\Delta\chi/\chi_0 = (\chi^{(2)}_{ijk} - \chi^{(2)}_{ijk,0})/\chi^{(2)}_{ijk,0}$, where $\chi^{(2)}_{ijk,0}$ refers to the tensor element $\chi^{(2)}_{ijk}$ at $t \leq 0$ ps. The error bars refer to the standard deviation resulting from the fit of the tensor elements. Inset: sketch of Ti atom and O octahedron with the indication of $O_\parallel$ (yellow spheres) and $O_\perp$ (red

spheres), and the softening of Ti-$O_\parallel$ bonds (dashed olive and cyan lines) with respect to the Ti-$O_\perp$ bonds (solid purple lines). **e, f** Relative change of SHG $\Delta I^p_{SHG}/I^p_{SHG,0}$ and reflectivity $\Delta R/R_0$ as a function of the delay $t$. $\Delta I^p_{SHG}/I^p_{SHG,0} = [I^p_{SHG}(t) - I^p_{SHG,0}]/I^p_{SHG,0}$, where $I^p_{SHG,0}$ refers to the SHG intensity at $t \leq 0$ ps. $\Delta R/R_0 = [R(t) - R_0]/R_0$, where $R_0$ refers to the reflectivity at $t \leq 0$ ps. The solid lines are fit curves to the data with fit parameters $\tau_0$, $\tau_1$, and $\tau_2$ reported in the text. The error bar of $\Delta I^p_{SHG}/I^p_{SHG,0}$ and $\Delta R/R_0$ data points are ≈ 8% and 2%, respectively. Since SHG is a nonlinear process, it is more significantly affected by fluctuations of the 800 nm probe laser intensity of ≈ 2%. However, the standard deviation of $\Delta I^p_{SHG}/I^p_{SHG,0}$ and $\Delta R/R_0$ at $t < 0$ ps are 0.6% and 0.06%, respectively.

time, we observe a fast increase in reflectivity by 7% within 350 fs, followed by two recovery times, $\tau^R_1 = 5.2 \pm 0.1$ ps and $\tau^R_2 = 29.8 \pm 0.5$ ps (Fig. 2f).

In both tr-SHG and tr-refl data, the time needed to reach the maximum relative change (≈ 350 fs) might be due to the thermalization of photoexcited electrons via electron-electron scattering. Subsequently, thermalized electrons, which are higher in the conduction band, move to the bottom of the conduction band, transferring energy to the phonon system, and recombining with holes in the valence band via electron-phonon scattering[4,40,45] or radiatively[46]. These processes are characterized by the recovery times $\tau_1$ and $\tau_2$. Both $\tau_1$ and $\tau_2$ recovery constants of $\Delta I^p_{SHG}/I^p_{SHG,0}$ are larger than those of $\Delta R/R_0$

because the dynamics of the spontaneous polarization (seen by tr-SHG) results from the convolution of the faster dynamics of photoexcited carriers (seen by tr-refl) and the slower dynamics of atoms.

To interpret the SHG intensity drop and the reflectivity increase, it is useful to express the magnitude of the spontaneous polarization as[22,47]: $P_s(t) = (1/V)\sum_i q_i(t)\Delta z_i(t)$, where $V$ is the volume of the unit cell, $q_i(t)$ is the local charge and $\Delta z_i(t)$ is the out-of-plane displacement of atom $i$ from the high symmetry position. The above-bandgap photoexcitation transfers electrons from the O 2p-derived valence band to the Ti 3d-derived conduction band of BTO. This charge transfer from O to Ti atoms reduces both the local negative charge at the O site ($q_O$) and the local positive charge at the Ti site ($q_{Ti}$). We attribute the increase in $R$ to

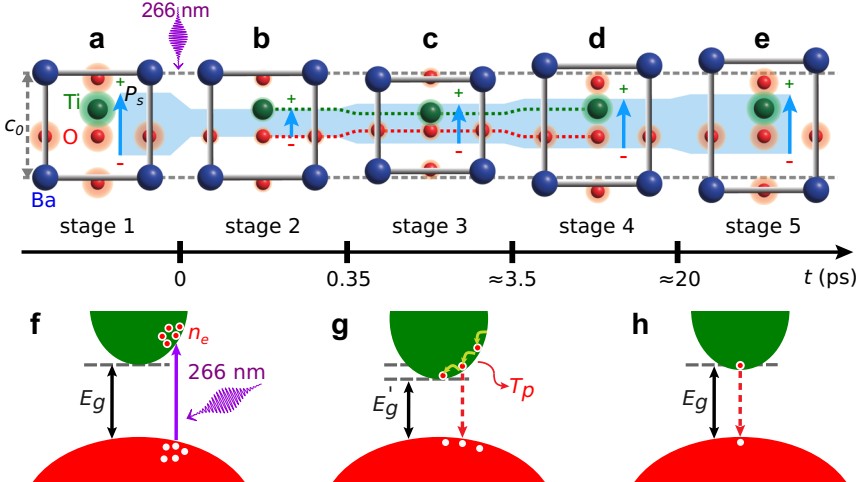

**Fig. 3 | Schematic model of photoinduced electron, polarization and lattice dynamics.** BTO unit cell, projected along the [100] direction, at the following time delays $t$: $t < 0$ ps, i.e., ground state (**a**), 0 ps $< t < 0.35$ ps (**b**), 0.35 ps $< t < 3.5$ ps (**c**), 3.5 ps $< t < 20$ ps (**d**), $\geq 20$ ps (**e**). The distance between the horizontal dashed gray lines indicates the out-of-plane lattice parameter at equilibrium $c_0$ in (**a**–**e**). The size of the green and red shaded areas around Ti and O atoms, is proportional to the corresponding positive and negative local charges, respectively. For clarity, the local charge of Ba atoms is omitted. The length of the light blue arrow and the height of the blue shaded area are proportional to the magnitude of the spontaneous polarization $P_s$. Sketch of BTO valence and conduction band in stage 2 (**f**), in stages 3 and 4 (**g**), and in stage 5 (**h**). The purple vertical arrow indicates the photoexcitation of electrons from the valence to the conduction band upon absorption of a 266 nm laser photon. $E_g = 3.4$ eV is the bandgap of the ground state, $E_g'$ is the bandgap of the excited state, modified by the deformation potential, with $E_g' - E_g \approx -3.2$ meV (Supplementary Note 3). The relaxation of the hot electrons to the bottom of the conduction band (green arrows) and the recombination with holes of the valence band (red dashed vertical arrow) induce an increase of $T_p$. **h** indicates the minor contribution of the deformation potential, resulting in a bandgap close to $E_g$, and the residual presence of electrons in the conduction band.

the increase in photoexcited carrier density $n_e$[38–40], which decreases $q_i$, while $P_s$ results from changes in both $q_i$ and $\Delta z_i$. This offers the opportunity to manipulate $P_s$ by modifying $q_i$, independently from the atomic displacements $\Delta z_i$. In fact, in our experiments, after 350 fs, before any significant atomic movement can occur, the increase in carrier density $n_e$ is responsible for the measured decrease in $P_s$ and increase in $R$, shown in Fig. 2e, f. This interpretation is corroborated by the increase of the maximum relative change of both $\Delta I^p_{SHG}/I^p_{SHG, 0}$ (negative, indicating decrease in polarization) and $\Delta R/R_0$ (positive, indicating increase in photoexcited carrier density) with pump fluence (Supplementary Fig. 10 and Supplementary Fig. 11). After a few tens of picoseconds, the longer recovery time of the polarization, as compared to the photoexcited carriers, is due to the contribution of the much slower structural dynamics, affecting $\Delta z_i$, characterized by a recovery time beyond the 40 ps timescale, as seen in Fig. 1d.

An interpretation of SHG intensity and reflectivity in terms of nonlinear SHG coefficient and electric susceptibility, equivalent to the one discussed above, is reported in Methods. Furthermore, we note that measuring current-voltage (I-V) curves is an alternative tool to measure photocurrents under illumination[48,49]. These measurements are typically performed under steady-state conditions and require a top electrode. Given the absence of top electrode in our sample, adding it would certainly change the interfacial properties of our sample and influence the measured photogenerated electrons. As an example, it has been shown that the photocurrent may vary by more than 2 orders of magnitude, depending on the top electrode employed[50]. Furthermore, modifying the top interface would also modify the BTO ferroelectric polarization, at least locally near the surface, and possibly also deeper in the BTO layer, as investigated experimentally[11,31,51] and theoretically[47]. Finally, adding a top electrode would also affect the strain propagation because of the different acoustic reflection coefficient at the BTO surface (Supplementary Table 1). Therefore, we probe the photoexcited carrier dynamics in our sample by tr-refl, as previously done in other works[38–40], and as typically done in ultrafast measurements[29].

## Discussion

We present here a unified picture in five stages of the electron, polarization, and lattice dynamics data presented above. Our observations are summarized in Fig. 3a–e, and the physical mechanisms involved are sketched in Fig. 3f–h. Before the arrival of the pump laser, the BTO is characterized by an out-of-plane polarization with given local charges at each atomic site (stage 1, Fig. 3a). Upon absorption of the pump laser at $t = 0$ ps, electrons move from the occupied O-derived valence band to the unoccupied Ti-derived conduction band (Fig. 3f). Within $\approx 350$ fs we observe the maximum increase in the photoexcited carrier density $n_e$, as indicated by the increase in $\Delta R/R_0$. This charge transfer reduces the local charges $q_i$ at Ti and O atoms, thereby decreasing the magnitude of the spontaneous polarization $P_s$, as indicated by the decrease in $\Delta I^p_{SHG}/I^p_{SHG, 0}$ (stage 2, Fig. 3b). A smaller $P_s$ is also consistent with a larger screening of long-range Coulomb interactions, which favor off-center atomic displacements and thus are responsible for the polar order. This effect is modeled by DFT calculations[52,53], showing that an increase in photoexcited carriers in the conduction band of BTO indeed tends to induce a phase transition from the ferroelectric to the paraelectric phase. These theoretical studies[52,53] investigated electron doping concentration per formula unit $n_{e/f.u.} \lesssim 0.2$, which corresponds precisely to the $n_{e/f.u.}$ range reached in our experiments (Supplementary Fig. 12). Moreover, TD-DFT calculations[32,33] predict that such photoinduced change of BTO electronic structure weakens more significantly Ti-O$_\parallel$ bonds, parallel to $\mathbf{P}_s$, as compared to Ti-O$_\perp$ bonds, perpendicular to $\mathbf{P}_s$. We demonstrate this effect experimentally by measuring a larger change of the out-of-plane tensor elements ($\chi_{zxx}, \chi_{zzz}$) as compared to the in-plane tensor element ($\chi_{xxz}$). In stage 2, in contrast to the maximum change in carrier density $n_e$ and polarization magnitude $P_s$, the lattice remains unperturbed: this marks the onset of the decoupling between polarization and strain. At these early time delays, the bulk photovoltaic effect (BPVE)[12] and the Schottky interface effect[11] could potentially play a role, but we explain in the following why they are not dominant in our experiments.

First, in photoexcited ferroelectric materials it is common to observe the BPVE, i.e., the generation of photovoltage under light illumination[54–56]. The BPVE occurs under two conditions: the presence of a noncentrosymmetrical crystal and the excitation of non-thermalized electrons[54]. In our system both conditions are satisfied shortly after $t = 0$ ps, i.e., before hot electron thermalization takes place (Fig. 3f). In BTO, if the light polarization is perpendicular to the spontaneous polarization $\mathbf{P}_s$, the induced photovoltage is parallel to $\mathbf{P}_s$[57,58]. This enhances the magnitude of the spontaneous polarization $P_s$ and induces, via the inverse piezoelectric effect, lattice expansion, similarly to what was observed in $BiFeO_3$[59]. However, in our study, although the polarization of the pump laser is perpendicular to $\mathbf{P}_s$ (Fig. 2a), within ≈ 350 fs, when electrons are nonthermalized, we see a decrease in $P_s$ (Fig. 2e). Hence, we conclude that, at this early timescale, BPVE is not a dominant effect in our experiments. In fact, this obser-vation can be explained by the wavelength dependence of the BPVE, which shows a maximum when the photon energy is close to $E_g$, as observed, e.g., in refs. [12,60]. In contrast, in BTO at $\lambda = 266$ nm the contribution of BPVE is negligible, as observed in experiments[57,58] and calculations[55].

Second, ferroelectric thin films grown on a metal substrate form a Schottky interface, where the electron-hole pairs generated by light absorption are separated by the built-in voltage of the Schottky bar-rier. The photoexcited carriers are most efficiently separated in the depletion region of width $w$, where the built-in field exists. At the BTO/SRO interface[61] the width of the Schottky interface is $w ≈ 5$ nm, similar to the one of other ferroelectrics, e.g., $Pb(Zr_{0.2}Ti_{0.8})O_3$/SRO interface[11,62]. Outside the depletion region, carrier recombination dominates because of the small diffusion length $L_d ≈ 2$ nm (Supple-mentary Note 6). Given the transmittance profile of the pump laser beam in our 34.5 nm thick BTO film (Supplementary Note 7 and Sup-plementary Fig. 13), only ≈ 5% of the pulse energy deposited in BTO is absorbed in the depletion region of the Schottky barrier and may contribute to the variation of the polarization[11]. Therefore, in our case the Schottky interface effect is expected to play a minor role.

In the 350 fs – 3.5 ps delay range (stage 3, Fig. 3c), we observe polarization and strain following opposite trends. In fact, while the displacement between Ti and the center of the O octahedron as well as the strain starts to decrease, $P_s$ starts to increase from its initial light-induced minimum. This is attributed to the start of recombination of photoexcited carriers, suggested by the incipient recovery in $\Delta R/R$. At this stage, we also observe lattice compression caused by the negative parameter $\partial E_g/\partial p$ via the deformation potential. Our observation is in line with theory[33,53,63,64] predicting a reduced bandgap $E_g$ due to the presence of photoinduced carrier density, leading to a reduced mag-nitude of the polarization $P_s$ (Fig. 3g). Moreover, the polarization trend is also consistent with the theoretical prediction[65] that photoexcited carriers screen dipole-dipole interaction, thereby reducing the overall polarization $P_s$ and, conversely, the progressive recombination of carriers in the conduction band result in a recovery of the polarization.

In the 3.5 – 7 ps delay range (stage 4, Fig. 3d), the relative dis-placement between Ti and O atoms tends to grow again as it starts to follow the increasing polarization, resulting in an increase of the out-of-plane strain. At the same time, the relaxation of photocarriers to the bottom of the conduction band and their recombination result in a further increase of the local charges, and an energy transfer to the phonon temperature (Fig. 3g), which, in turn, leads to lattice expan-sion. The latter effects persist up to $t ≈ 20$ ps.

At $t ≈ 20$ ps (stage 5, Fig. 3e), together with a further relaxation of the electronic system toward equilibrium, we observe a saturation of the BTO average strain. This results in a metastable state with a slightly reduced $P_s$ and a significantly increased out-of-plane strain with respect to the ground state. The reduced $P_s$ is attributed to the pre-sence of residual photoexcited carriers in the conduction band (Fig. 3h), while the increased out-of-plane strain is due to the

thermoelastic contribution. We exclude the depolarization field screening as the main driving effect for the transient tensile strain seen in our experiments, because it would lead to the typical saturation of the strain with increasing pump fluence[66]. For example, in $PbTiO_3$[12] the metastable lattice strain reaches saturation with 266 nm laser fluences of 0.01 mJ cm$^{-2}$. Conversely, in our study, we employ two orders of magnitude larger pump fluence, with similar laser pulse duration, and still the metastable strain scales approximately linearly with the pump fluence (Fig. 1d and Supplementary Fig. 4). In addition, on the ≈ 20 ps timescale, the deformation potential plays a minor role (Supplemen-tary Fig. 6), thus its influence on the bandgap is expected to be mar-ginal. Interestingly, although the out-of-plane lattice parameter is larger than in the ground state ($c_0$), the polarization $P_s$ is still smaller compared to equilibrium. This underscores the persistence of the decoupling between polarization and strain. Quantitatively, we esti-mate that in stage 5 the contribution of photoexcited electrons to $P_s$ has ≈ 10% larger magnitude than the structural contribution from the strain $\eta$ (Supplementary Note 8).

In summary, we demonstrate that, with an above-bandgap laser excitation, an ultrafast decoupling of spontaneous polarization and strain can be achieved and measured. The key difference to previous studies is that we measure both the polarization and the strain, and thus we are able to observe the ultrafast decoupling of polarization and strain taking place already within 350 fs and lasting at least for several tens of picoseconds. Another important difference to previous reports is that we employ a larger peak power intensity of the pump laser (39 GW cm$^{-2}$, see Supplementary Note 4) with excitation energy above the bandgap. This is important to excite a sufficient number of electrons from the O 2p-derived valence band to the Ti 3d-derived conduction band, reaching ≈ 0.2 e/f.u. (Supplementary Fig. 12), and to induce the change in polarization, as predicted by theory[52,53]. We assign the ultrafast decoupling of polarization and strain to the dominant contribution of photoexcited electrons in determining the magnitude of the spontaneous polarization when the system is out of equilibrium, and show that for an accurate description and fundamental under-standing of a ferroelectric material in a photoexcited state, it is essential to combine multiple techniques that address the dynamical evolution of the various degrees of freedom, i.e., electrons, ferro-electric polarization, and lattice. The decoupling of the polarization from the strain offers the opportunity to change paradigm from strain engineering[27,28,67] to light-induced polarization engineering, thereby lifting the constraint of selecting among a limited number of substrates[24] or designing freestanding membranes[26] to tune the spontaneous polarization. Moreover, by softening the Ti-O$_\parallel$ bonds, we bring BTO to an excited state, where it could be further modified by THz light to achieve stable and reversible polarization switching at lower fluences than otherwise needed when starting from the ground state[21]. Finally, the transient and reversible control of the spontaneous polarization, shown in this study, offers a pathway to control by light both electric and magnetic degrees of freedom in multiferroic materials.

## Methods

### Sample preparation and characterization

The epitaxial bilayer BTO/SRO was grown using pulsed laser deposi-tion on a GSO substrate, with (110) orientation according to the orthorhombic notation. The ceramic targets of SRO and BTO were 8 cm away from the substrate and ablated using a KrF excimer laser ($\lambda = 248$ nm, fluence 5.4 J cm$^{-2}$, 2 Hz repetition rate). The deposition of SRO and BTO layers was conducted in $O_2$ atmosphere with pressure $pO_2 = 100$ mTorr and deposition temperatures of 908 K and 973 K, respectively. Sample cooling with the rate of 3 K min$^{-1}$ was conducted in the environment of saturated $O_2$ ($pO_2 = 10^4$ mTorr) to prevent the formation of oxygen vacancies. The thicknesses of BTO and SRO lay-ers, $d_{BTO} = 34.5$ nm and $d_{SRO} = 47$ nm are extracted from a $\theta$–$2\theta$ scan of

the as-grown sample around the (002) reflections (Supplementary Fig. 14), while the GSO substrate is 0.5 mm thick. We performed $\theta$-$2\theta$ scans as a function of sample temperature (Supplementary Fig. 15) to determine the thermal expansion coefficients of BTO above $T_c = 400\,°C$, of SRO and GSO in the temperature range $35\,°C < T < 700\,°C$. We determine the out-of-plane lattice parameters $c_{BTO} = 4.074$ Å, $c_{SRO} = 3.934$ Å, and $c_{GSO} = 3.964$ Å by means of the reciprocal space map (RSM) shown in Supplementary Fig. 16. The in-plane lattice parameter $a = 3.970$ Å, common to BTO, SRO and GSO, indicates that both BTO and SRO thin films are coherently strained to the substrate. The in-plane compressive strain of $-0.55\%$ imposed to BTO by the substrate is calculated by comparing in-plane lattice parameter $a$ with the respective bulk value $a_{b,BTO} = 3.992$ Å[68], according to $(a - a_{b,BTO})/a_{b,BTO}$. The absence of satellite peaks in RSM data (Supplementary Fig. 16) suggests the existence of a BTO monodomain. The single-crystalline nature of the BTO film, with no visible stress or dislocations, is demonstrated by scanning transmission electron microscopy (STEM) data (Supplementary Fig. 17a and Supplementary Note 9). Both SRO/GSO and BTO/SRO interfaces are of high-quality. The former is atomically sharp (Supplementary Fig. 17b), and the latter has steps not exceeding one unit cell with interdiffusion below the detection limit (Supplementary Fig. 17c–g). The single-domain nature of the BTO film is further demonstrated by means of piezoresponse force microscopy (PFM, Supplementary Fig. 18), which provides the spontaneous polarization of the as-grown sample $\mathbf{P}_s$ pointing upward (toward the surface). Furthermore, the independence of both SHG polar plots of $I^p_{SHG}(\varphi)$ and $I^s_{SHG}(\varphi)$ from the azimuthal angle $\gamma$ (Fig. 2a) confirms the out-of-plane nature of the spontaneous polarization of our BTO sample, and the absence of a net in-plane component of the polarization (Supplementary Fig. 19).

## Time-resolved X-ray diffraction

Time-resolved X-ray diffraction experiments were performed in the X-ray resonant diffraction chamber at the Spectroscopy and Coherent Scattering instrument (SCS) of the European X-Ray Free-Electron Laser Facility (EuXFEL), using an optical laser (OL) as pump and the XFEL as probe. The XFEL pulse pattern consisted of pulse trains at a repetition rate of 10 Hz, with 35 pulses per train at the intratrain repetition rate of 113 kHz. The full width at half maximum (FWHM) of the XFEL spectrum was 11.7 eV. To reduce the energy bandwidth, the XFEL beam was monochromatized using a variable line spacing grating with 50 lines/mm in the first diffraction order and exit slits with a gap of 100 µm, providing an energy resolution of $\approx 650$ meV. The nominal pulse duration of the XFEL pulses was $\approx 25$ fs, with pulse stretching at the monochromator grating of $\approx 10$ fs (FWHM). The XFEL pulses, with initial energy of 1.5 mJ per pulse, were then attenuated by transmission through a gas attenuator (GATT), consisting of a volume containing $N_2$ gas at a variable pressure. In order to prevent detector saturation, the transmission of the GATT was set to have $\approx 15$ nJ per pulse at the sample. The XFEL pulse energy was measured by an X-ray gas monitor detector (XGM), located $\approx 7$ m upstream of the sample, just before the Kirkpatrick-Baez (KB) mirrors. The latter were used to focus the XFEL beam at the sample to a spot size of $w^{XFEL}_x \times w^{XFEL}_y = 140 \times 100\,µm^2$ (determined by knife edge scans), where $w$ is defined as the distance from the beam axis where the intensity drops to $1/e^2$ of the value on the beam axis. The angle of incidence (defined from the sample surface) of XFEL and OL beams at the sample was 86° and 85°, respectively. The intensity of each XFEL pulse diffracted by the sample was measured by a Si avalanche photodiode (APD, model SAR3000G1X, Laser Components), converted to a voltage pulse and digitized. To prevent the pump laser intensity from reaching the APD, the latter was equipped with a filter made of 400 nm Ti, deposited on 200 nm polyimide.

The pump laser had a central wavelength of 266 nm, the same pulse pattern as the XFEL with pulse duration of 70 fs, and beam size $w^{OL}_x \times w^{OL}_y = 330 \times 240\,µm^2$ determined by knife edge scans. The

266 nm pump laser incident on our sample is partially reflected at the BTO surface (18%), it is mostly absorbed in the BTO thin film (70%), and to a smaller extent it is absorbed in the SRO layer (12%), as shown in Supplementary Fig. 13 and explained in Supplementary Note 7. As a result, GSO has no contribution to the optical response of the ferroelectric thin film. The incident pump laser fluence at the sample was 2.7 mJ cm$^{-2}$, and we verified that diffraction curves $I_{XRD}(E_\nu)$ measured at negative delays $t$ coincide with those measured without the pump laser (Supplementary Fig. 20). This confirms the reversibility of the pump effect induced by UV laser light. The temporal overlap between XFEL and OL was determined as detailed in Supplementary Fig. 21. The diffracted intensity of the XFEL pulses in a train was averaged and the corresponding time delay of the OL was corrected by the respective bunch arrival monitor (BAM) value (Supplementary Fig. 22), obtaining a time resolution $\Delta t \approx 90$ fs (Supplementary Note 10). Data measured with incident pump laser fluence of 1.4 mJ cm$^{-2}$ are reported in Supplementary Note 2 and Supplementary Note 4.

In general, two kinds of tr-XRD experiments were performed: photon energy scans at a fixed pump-probe delay $t$ (Fig. 1b and Supplementary Fig. 1), and time delay scans at a fixed photon energy $E_\nu = 1525$ eV (Fig. 1c and Supplementary Fig. 21). The energy scans were carried out by a simultaneous movement of the monochromator grating and the undulators gap, such to have always the peak of the XFEL spectrum at the desired photon energy. From the energy scans at different delay $t$, the out-of-plane lattice parameters $c(t)$ and $c_O$ are calculated as the center-of-mass of the BTO diffraction intensity $I_{XRD}(t)$ and refer to the average $c$ over $d_{BTO}$ ($\bar{c}$). Specifically, the BTO average out-of-plane parameters $\bar{c}$ is calculated from the corresponding (001) Bragg peaks using the Bragg condition $\bar{c} = (12400\text{ eV Å})/(2\bar{E}_\nu \sin\theta)$, where $\bar{E}_\nu$ is the average of energy values, around the (001) BTO peak, weighted by $I_{XRD}(E_\nu)$. Energy scans at different pump-probe delays $t$ over the photon energy range covering also GSO (001) and SRO (001) Bragg peaks are reported in Supplementary Fig. 23. The comparison of simulated and experimental diffraction curves before the arrival of the pump laser is reported in Supplementary Fig. 24.

## Time-resolved SHG and reflectivity

Time-resolved Second Harmonic Generation and time-resolved reflectivity experiments were performed at the SCS instrument of the EuXFEL using the same optical laser employed for tr-XRD experiments and the same pulse pattern. A sketch of the setup is shown in Fig. 2a. The 800 nm probe beam at frequency $\omega$ (red arrow), with electric field $\mathbf{E}(\omega)$, impinges on the sample at angle $\theta = 50°$ (defined from the normal to the surface) with polarization defined by the angle $\varphi$ and varied by rotating a half waveplate. The angle $\varphi = 0°$ [$\varphi = 90°$] refers to $p$ [$s$] polarized light. The 266 nm pump beam at frequency $3\omega$ (purple arrow) impinges on the sample at normal incidence with $p$ polarization. The 800 nm probe beam is then reflected by the sample and a dichroic mirror before reaching a Si photodiode. The latter is used to measure tr-refl of our sample. The 400 nm beam at frequency $2\omega$ (blue arrow) is the SHG signal generated in the BTO sample (Supplementary Fig. 25). This SHG beam is transmitted through the dichroic mirror and Glan polarizer, which is set to select either the $p$ or $s$ component of the electric field ($E_p(2\omega)$ or $E_s(2\omega)$), corresponding to the $p$-out or $s$-out configuration, respectively. Finally, the SHG beam is filtered by a 400 nm bandpass filter before reaching a photomultiplier (model H10721-210-Y004, Hamamatsu). The azimuthal rotation of the sample around the $z$ axis is defined by the angle $\gamma$. Both Si photodiode and photomultiplier measure respectively the reflectivity and SHG signal of each OL pulse in the train. The pulse duration of the 800 nm probe and 266 nm pump beams were $\approx 50$ fs and 70 fs, providing a time resolution of $\approx 90$ fs. The beam sizes of 800 nm and 266 nm beams, determined by knife edge scans, were $w^{800\,nm}_x \times w^{800\,nm}_y = 55 \times 46\,µm^2$ and $w^{266\,nm}_x \times w^{266\,nm}_y = 165 \times 311\,µm^2$, respectively. The penetration

depths of 800 nm, 400 nm and 266 nm laser beams in BTO are reported in Supplementary Note 7B. While the fluence of the 800 nm probe beam was kept at 1.3 mJ cm⁻², the fluence of the 266 nm pump beam was set to 2.7 mJ cm⁻² for the data displayed in Fig. 2. tr-SHG and tr-refl data with fluences between 1.4 mJ cm⁻² and 12.3 mJ cm⁻² are reported in Supplementary Fig. 10 and Supplementary Fig. 11.

## Fit curves of SHG data

In general, given the incoming electric field with components $E_j(\omega)$ and $E_k(\omega)$ at frequency $\omega$ along $j$ and $k$ directions, the electric dipole polarization induced in the material at frequency $2\omega$ along the direction $i$ is $P_i(2\omega) = \chi^{(2)}_{ijk} E_j(\omega) E_k(\omega)$, where $\chi^{(2)}_{ijk}$ is the second-order susceptibility, i.e., a third rank tensor reflecting the symmetry of the material. Each direction $i, j, k$ can be $x$, or $y$, or $z$ directions (Fig. 2a). By selecting $p$ or $s$ polarization of the SHG beam, we measure $I^p_{SHG}(\varphi)$ or $I^s_{SHG}(\varphi)$, which, for a BTO single crystal with *4mm* point group symmetry, can be expressed as[11,42]:

$$I^p_{SHG}(\varphi) \propto \left( \chi_{zxx} \sin\theta \cos\varphi^2 + \left(2\chi_{xxz} \cot\theta^2 + \chi_{zxx} \cot\theta^2 + \chi_{zzz}\right) \sin\theta^3 \sin\varphi^2 \right)^2,$$

(1)

$$I^s_{SHG}(\varphi) \propto \left(2\chi_{xxz} \sin\theta \sin\varphi \cos\varphi\right)^2.$$

(2)

The resulting ratios of the tensor elements in Fig. 2b, c, $\chi^{(2)}_{zzz}/\chi^{(2)}_{zxx} = 3.7$ and $\chi^{(2)}_{zzz}/\chi^{(2)}_{xxz} = 6.5$, reflect a thin film under tensile out-of-plane stress[69]. The good quality of the fit curves in Fig. 2b, c confirms that our BTO thin film has *4mm* point group symmetry. The minor discrepancies between data and fit model might be due to the fact that our BTO thin film is coherently strained to the substrate, and this leads to the appearance of additional minor nonzero tensor elements[70].

## Interpretation of SHG intensity and reflectivity

In the section "Results" we discuss SHG intensity and reflectivity as proportional to the magnitude of the spontaneous polarization $P_s$ and the photoexcited carrier density $n_e$, respectively. In the following, we show an equivalent interpretation of SHG intensity and reflectivity in terms of nonlinear SHG coefficient and electric susceptibility.

In general, the material permittivity $\varepsilon$ relates the applied electric field **E** to the electric displacement field **D** according to $\mathbf{D} = \varepsilon\mathbf{E} = \varepsilon_0\mathbf{E} + \mathbf{P}$, where $\varepsilon_0$ is the vacuum permittivity and **P** is the induced electrical polarization. Following the notation in ref. 42, **P** can be expressed as:

$$\mathbf{P}(t) = \mathbf{P}_0 + \varepsilon_0\chi_e\mathbf{E}(t) + \chi^{(2)}\mathbf{E}(t)^2,$$

(3)

where $\mathbf{P}_0$ is a time-independent constant polarization. The second term in eq. (3) depends linearly on $\mathbf{E}(t)$ and induces electric dipole oscillations at $\omega$, i.e., the same oscillation frequency as the incident electric field of light, thus it is responsible for the optical reflectivity. In eq. (3), $\chi_e = n_\omega^2 - 1$ is the electric susceptibility of the material and $n_\omega$ is the refractive index at $\omega$. The third term in eq. (3) has a quadratic dependence on $\mathbf{E}(t)$ and induces electric dipole oscillation at $2\omega$, twice the frequency of the incident light, thus it is responsible for the second harmonic generation process. In particular, if we ignore the tensor form, the nonlinear SHG coefficient can be expressed as[42]:

$$\chi^{(2)} \propto A\chi_e^2(\omega)\chi_e(2\omega),$$

(4)

where $\chi_e(2\omega) = n_{2\omega}^2 - 1$, with $n_{2\omega}$ being the refractive index at $2\omega$, and $A$ is a structural parameter related to the atomic displacements $\Delta z_i$ from the high symmetry positions. In materials with a center of inversion symmetry, $A = 0$, $\chi^{(2)} = 0$, and the second order term of the induced polarization vanishes. Therefore, tr-SHG measurements are sensitive to both electronic (via $\chi_e(\omega)$ and $\chi_e(2\omega)$) and structural (via $A$) changes

in the material. At the same time, tr-refl measures changes in the electric susceptibility $\chi_e(\omega)$, which, in our work, result from the photoexcitation of electrons from the O 2p-derived valence band to the Ti 3d-derived conduction band, while structural changes are monitored by tr-XRD. The SHG intensity is proportional to $|\chi^{(2)}|^2$, and thus to a combination of both electronic and structural contributions. The time-evolution of $\Delta I^p_{SHG}/I^p_{SHG,0}$ and $\Delta R/R_0$ (Fig. 2e, f) indicates that $\chi^{(2)}$ is mostly determined by the changes in electric susceptibility $\chi_e(\omega)$, and to a smaller extent to the structural changes in $A$. This is equivalent to say that changes in the magnitude of the spontaneous polarization $P_s$ are mainly determined by the photoexcited electrons and in minor part to the structural dynamics of the lattice, as discussed in the section "Photoinduced ferroelectric polarization and electron dynamics (Results)".

## Reporting summary

Further information on research design is available in the Nature Portfolio Reporting Summary linked to this article.

## Data availability

Time-resolved X-ray diffraction raw data recorded at the European XFEL are available at https://doi.org/10.22003/XFEL.EU-DATA-003481-00. The data supporting the findings of the study are available in Figshare at https://doi.org/10.6084/m9.figshare.28381727.v2.

## Code availability

Codes generated during the current study to fit the time-dependent strain profiles, and simulate the time-dependent strain profiles and X-ray diffraction curves are available at https://doi.org/10.24433/CO.9206382.v1.

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

## Acknowledgements

We acknowledge the European XFEL in Schenefeld, Germany, for provision of the XFEL beamtime at the SCS scientific instrument and would like to thank the staff for their assistance. D.P. acknowledges funding from 'la Caixa' Foundation fellowship (ID 100010434) and the Spanish Ministry of Industry, Economy and Competitiveness (MINECO), grant no PID2019-109931GB-I00. The ICN2 is funded by the CERCA programme/Generalitat de Catalunya and by the Severo Ochoa Centres of Excellence Programme, funded by the Spanish Research Agency (AEI, CEX2021-001214-S). E.M.U., D.K., and M.D. acknowledge support from the Zukunftsfonds Steiermark for infrastructure funding (project "ASTEM Upgrade") and from the Wirtschaftskammer Steiermark (WKO Stmk.) for providing additional funds for maintenance. T.C.A. acknowledges funding from the Heisenberg Resonant Inelastic X-ray Scattering (hRIXS) Consortium. The work by G.Merz. was jointly supported by Politecnico di Milano and European X-ray Free Electron Laser Facility GmbH. G.C. acknowledges financial support from the Catalan government (grant number 2021 SGR 0129), and from the Spanish Research Agency (Agencia Estatal de Investigación), project number PID2023-148673NB-I00. We thank D. Hickin for support with the automation of tr-SHG measurements. We thank A. Reich and J.T. Delitz for the design of mechanical components used in tr-XRD experiments. We are grateful to M. Altarelli for careful reading of the manuscript.

## Author contributions

G.Merc., with input from L.P.H., G.C., I.A.V., J.Z. and T.L.L., conceived the experiments. G.C. and I.S., with input from G.Merc., designed the sample. J.M.C.R. manufactured the sample and D.P. characterized it with $\theta$–$2\theta$ scan, RSM and PFM. L.P.H., D.P., G.N.H., R.C., L.M., M.T., S.G., T.C.A., G.Merz., S.P., J.Sc., Z.Y., I.A.V. and G.Merc. performed tr-XRD experiments. C.C. developed an online analysis tool to visualize tr-XRD data in real time. G.Merc. and L.P.H. performed tr-SHG and tr-refl experiments. J.Sa. performed $\theta$–$2\theta$ measurements at different sample temperatures. E.M.U. performed STEM measurements with the supervision of D.K., and M.D. prepared the lamella for STEM measurements. L.P.H. and G.Merc. analyzed all the data. L.P.H. modeled tr-XRD and tr-SHG data. G.Merc., A.S., K.R. and I.A.V. supervised the project. G.Merc. wrote the manuscript, drafted by L.P.H., and with input from all authors. All authors provided critical feedback and helped shape the research, analysis and manuscript.

## Funding

## Competing interests

The authors declare no competing interests.
