## [Transparent Peer Review file · Nature Communications]

Ultrafast decoupling of polarization and strain in ferroelectric BaTiO₃

Corresponding Author: Dr Giuseppe Mercurio

Version 0:

Reviewer comments:

Reviewer #1

(Remarks to the Author)

Understanding the interplay between lattice structure, polarization, and electrons is crucial for the optical control in ferroelectrics. In this work, the authors induce and measure an ultrafast decoupling between polarization and strain within 350 fs by above-bandgap laser excitation in ferroelectric BaTiO₃, which further show that the unbalanced ferroelectric polarization is mainly determined by the photoexcited electrons rather than the strain. Effectively demonstrate that ultrafast decoupling of spontaneous polarization and strain can be achieved and detection using above-bandgap laser excitation. Therefore, the light-induced transient and reversible control of the ferroelectric polarization are novel topic with advanced experimental methods. I recommend the publication of this article in Nature Communications after revisions.

1. In my opinion, the Figures in the manuscript should be modified appropriately because they are not consistent. Firstly, the text size of the illustrations in all Figures is not uniform and varies greatly, such as Figure 1. Secondly, the serial numbers (such as a, b, c, d) of different pictures are different in size. For example, Figure 1 and Figure 3 are much larger than Figure 2. Thirdly, the font formats used in the Figures are inconsistent, such as in Figures 2a and 2b.
2. In the laser excitation experiment, light energy may cause thermal effects, thereby generating strain and affecting polarization. Is there any consideration of the thermal effect in this system? How did the authors eliminate it out?
3. In this work, XRD measurement is to confirm that the XRD peak shift is mainly due to c-axis strain. Can you further discuss the possible contribution of the substrate to the optical response of ferroelectric films? Is it necessary to do comparative tests on other substrates?
4. The author mentioned DFT calculations several times in the paper (page 5, line 140 and page 7, lines 168 and 170), which involve the key Ti-O bond. Can some calculations be added to the paper to compare and verify the experimental results? This will significantly improve the quality of the article.
5. The paper points out that photogenerated electrons complete the initial polarization change within 350 fs, while the time scale of polarization recovery is significantly longer than the recovery time of reflectivity change. The authors do not seem to discuss the specific contributions of different scattering mechanisms in polarization recovery, and additional discussion is recommended.
6. The I-V curves of ferroelectrics are important to discuss the photogenerated electrons. Thus, we suggest to add this results in the revised manuscript. Some literatures have discussed this issue, such as
-Journal of Applied Physics 132 (2022) 224106.
-Journal of the European Ceramic Society 44 (2024) 5752–5764.
7. The authors mentioned that the Schottky interface effect can be ignored in this work. It is better to provide additional calculation or experimental data to support this conclusion. Provide additional analysis or literature support to enhance the persuasiveness.
8. The author cited and compared many previous studies in analyzing the mechanism, but has not yet clearly explained the key differences between this study and existing reported. It is recommended to clarify the highlights of the article in the revised manuscript.

(Remarks on code availability)

No

Reviewer #2

(Remarks to the Author)

This study investigates the ultrafast decoupling of polarization and strain in the ferroelectric material BaTiO₃ (BTO) under above-bandgap UV light excitation. Using time-resolved X-ray diffraction (tr-XRD) and second-harmonic generation (tr-SHG), the researchers examine the separate dynamics of the lattice (out-of-plane strain), spontaneous polarization, and photoexcited carriers. They show that the decoupling occurs within 350 fs after excitation, attributed primarily to the effect of photoexcited electrons that influence the polarization, while the strain response is slower, dictated by the lattice dynamics. The results challenge the assumption that polarization and strain are always coupled in ferroelectrics, especially out of equilibrium, and demonstrate the potential for using light to manipulate the polarization independently of strain.

1. The authors should supplement their analysis with SAED or STEM data to directly confirm the single-domain nature of the BTO films. This would strengthen the argument and address potential ambiguities in the current dataset.
 2. The manuscript describes the use of In Situ X-ray Diffraction (IXRD) and optical methods to analyze photoinduced lattice strain. However, no Transmission Electron Microscopy (TEM) data is provided to directly observe atomic displacements, such as changes in Ti-O bond lengths. The authors should include TEM measurements to directly observe Ti-O bond length changes and validate the XRD-derived conclusions about photoinduced lattice strain.
 3. The manuscript does not address the potential influence of the interface structure between BTO/SRO/GSO heterostructures on the transfer of photoinduced strain. Features such as dislocations or stress concentrations at the interfaces could significantly affect how strain propagates through the system. Understanding these interface characteristics is critical for interpreting the observed photoinduced strain behavior.
- If these aspects are adequately addressed, the paper could indeed be suitable for publication in Nature Communications.

(Remarks on code availability)

Reviewer #3

(Remarks to the Author)

This paper investigates the ultrafast decoupling of polarization and strain in ferroelectric BaTiO₃ upon above-bandgap laser excitation. Through a combination of tr-XRD, tr-SHG, and tr-refl, the authors observe and measure the dynamics of lattice parameters, spontaneous polarization, and photoexcited carrier density. The study reveals that photoexcited electrons play a dominant role in determining the spontaneous polarization out of equilibrium, leading to an ultrafast decoupling between polarization and strain. The paper presents novel findings and employs advanced experimental techniques, making it suitable for publication in Nature Communications, provided the following issues are addressed:

1. The format of the formula in line 152 is misleading. The authors wrote it as $P_s(t) = 1/V \sum_i q_i(t) \Delta z_i(t)$, as the summation appears to be in the denominator. This should be clarified to ensure the summation is correctly placed in the numerator. Additionally, the interpretation of the SHG intensity drop and reflectivity increase based on this formula may need reconsideration.
2. The authors attempt to directly correlate the reduction in SHG intensity with the decrease of ferroelectric polarization. However, while the measured SHG intensity is proportional to the square of $\chi_{ijk}^{(2)}$, its magnitude is also influenced by the material's permittivity. The observed changes in tr-refl also indicate alterations in the dielectric characteristics of BTO. How can the authors exclude the relationship between the reduction in SHG signal intensity and the changes in dielectric properties? This point should be addressed in the discussion.
3. Please ensure that the figures and sections in the Supplementary Material are arranged in the order they are referenced in the main text. This will improve the coherence and ease of reference for readers.
4. The authors should clarify why charge transfer leads to a reduction in the Born effective charge. This would enhance understanding for a broader audience.
5. In line 203, the phrase "...to the top of the conduction band..." should be "... to the bottom of the conduction band..."
6. References related to strain manipulating ferroelectric polarization (DOI: 10.1002/adv.202417165 and DOI: 10.1002/adv.202307571) should be included to enrich the introduction part.

(Remarks on code availability)

Version 1:

Reviewer comments:

Reviewer #1

(Remarks to the Author)

The authors have made very detailed responses to the comments of all reviewers. More importantly, their responses are serious and persuasive, so the manuscript was greatly improved. I think the article is acceptable for publication and does not need further modification.

(Remarks on code availability)

Reviewer #2

(Remarks to the Author)

In the revised manuscript and response letter, the authors have added additional data and discussion to address the reviewers' questions. The work can meet the criteria of NC journal and the current version can be accepted.

(Remarks on code availability)

Reviewer #3

(Remarks to the Author)

The authors have addressed the issues raised by myself and the other reviewers. In its current form, I believe the manuscript now meets the publication standard of Nature Communications. Only one minor note remains: '4mm' should be italicized.

(Remarks on code availability)

Point-by point response to the reviewers

Below, we separately address the comments from the Reviewers marked in *black-italic* with our responses marked in **blue**. Portions of the manuscript reported here are enclosed in a box, with the text that remains unchanged marked in black and the changes in the manuscript marked in **red**.

Reviewer #1 (Remarks to the Author):

Understanding the interplay between lattice structure, polarization, and electrons is crucial for the optical control in ferroelectrics. In this work, the authors induce and measure an ultrafast decoupling between polarization and strain within 350 fs by above-bandgap laser excitation in ferroelectric BaTiO₃, which further show that the unbalanced ferroelectric polarization is mainly determined by the photoexcited electrons rather than the strain. Effectively demonstrate that ultrafast decoupling of spontaneous polarization and strain can be achieved and detection using above-bandgap laser excitation. Therefore, the light-induced transient and reversible control of the ferroelectric polarization are novel topic with advanced experimental methods. I recommend the publication of this article in Nature Communications after revisions.

Comment: 1. In my opinion, the Figures in the manuscript should be modified appropriately because they are not consistent. Firstly, the text size of the illustrations in all Figures is not uniform and varies greatly, such as Figure 1. Secondly, the serial numbers (such as a, b, c, d) of different pictures are different in size. For example, Figure 1 and Figure 3 are much larger than Figure 2. Thirdly, the font formats used in the Figures are inconsistent, such as in Figures 2a and 2b.

Answer: We thank the Reviewer for the generally positive assessment of our work. Regarding the inconsistencies pointed out by the Review in Comment 1, the font formats of all Figures in the revised manuscript are now uniform and consistent.

Comment: 2. In the laser excitation experiment, light energy may cause thermal effects, thereby generating strain and affecting polarization. Is there any consideration of the thermal effect in this system? How did the authors eliminate it out?

Answer: In our experiments, given the relatively high incident peak power intensity of 39 GW cm⁻², and the pump photon energy ~1.2 eV above the BTO bandgap, we take thermal effects into account. In fact, we developed a model based on the well-known two-temperature model (2TM) and one-dimensional lattice strain wave equation, in order to describe the time-evolution of the out-of-plane strain (Figure 1d). In a nutshell, after absorption of the pump laser, the thermalization of photoexcited electrons increases the electron temperature and changes the electronic system, which can be modelled by a reduction of the bandgap and thus initial lattice contraction. Subsequently, photoexcited electrons transfer part of their excess energy to the phonon system, thereby increasing the lattice temperature and inducing lattice expansion. The

details of the model and its results are described in great detail in Supplementary Notes 1,2, and 3.

Following the question of the Reviewer, in the revised manuscript, in the last paragraph of the Section 'Photoinduced structural dynamics' at page 4-5, we explicitly write that thermal effects are taken into account and extend the description of the 2TM with further references to the Supplementary Information as shown below:

The model fitting $\eta(t)$ data in Figure 1d is presented in the following. When a photon with energy $E > E_g$ is absorbed in BTO, electrons are photoexcited from the O 2p-derived valence band to the Ti 3d-derived conduction band [32–34]. The thermalization of photoexcited electrons leads to an increase in the electron temperature (T_e), and to changes in the electronic system that can be modeled by the variation of the bandgap as a function of the electronic pressure ($\partial E_g/\partial p$) [4]. In turn, a modified electron system affects the interatomic potential, resulting in atomic motions and contributing to the deformation potential stress $\sigma_{DP}(T_e, \partial E_g/\partial p)$. Subsequently, photoexcited electrons transfer part of their excess energy ($E - E_g$) to the phonon system via electron-phonon scattering, increasing the phonon temperature (T_p) on the picosecond timescale. This, in turn, induces a lattice expansion dependent on the BTO thermal expansion coefficient (β), and contributes to the thermoelastic stress $\sigma_{TE}(T_p, \beta)$. The total stress [4, 35] $\sigma = \rho v^2 \eta + \sigma_{DP} + \sigma_{TE}$ generates a strain wave $\eta(z, t)$ that propagates through the material of mass density ρ at the longitudinal speed of sound v . Given the incident peak power intensity of 39 GWcm^{-2} and other known sample parameters (Table S1), we solve analytically the two-temperature model (2TM, Supplementary Note 1) and the lattice strain wave equation (Supplementary Note 2) to obtain the strain $\eta(z, t)$. The 2TM describes electron and phonon temperatures, $T_e(z, t)$ and $T_p(z, t)$, upon absorption of a laser pulse in our sample, thereby accounting for thermal effects (Figure S3). Finally, we calculate the strain $\eta(t)$, averaged over d_{BTO} , and obtain an accurate fit of the experimental data in Figure 1d. A similarly good fit of $\eta(t)$ data is obtained for incident pump fluence of 1.4 mJcm^{-2} (Figure S4). The main outcome of our fit model is a negative $\partial E_g/\partial p$ of the order of $\approx 10^{-3} \text{ eVGP}^{-1}$ (Table S2), in agreement with first-principles calculations [36], with a resulting bandgap decrease of about 3.2 meV (Supplementary Note 3). The negative $\partial E_g/\partial p$ causes lattice compression along the out-of-plane direction within the first $\approx 3\text{ps}$, when the negative σ_{DP} dominates over σ_{TE} (inset of Figure 1d). Conversely, at larger time delays ($t > 3\text{ps}$), T_p increases (Figure S5) and the thermoelastic term becomes the dominant one, leading to an increase of the average out-of-plane strain $\eta(t)$ (Figure 1d and Figure S6). The calculations of lattice temperature, out-of-plane strain and diffraction curves as a function of delay t and distance z from the BTO surface are reported in Supplementary Note 4. The validity of the model employed to fit the strain $\eta(z,t)$ data is further corroborated by the good agreement between the experimental BTO (001) diffraction peaks measured at different pump-probe delays t and the corresponding calculated diffraction curves, based on the strain profiles as a function of delay t and distance z from the BTO surface (Figure S7 and Figure S8).

Furthermore, we demonstrate experimentally that to reduce thermal effects it is sufficient to employ a lower pump fluence, for example 1.4 mJcm^{-2} , instead of 2.7 mJcm^{-2} , as shown in Figure S3. However, as a consequence, changes in photoinduced strain (Figure S4), polarization and electron dynamics (Figures S10 and S11) decrease accordingly.

Comment: 3. In this work, XRD measurement is to confirm that the XRD peak shift is mainly due to *c*-axis strain. Can you further discuss the possible contribution of the substrate to the optical response of ferroelectric films? Is it necessary to do comparative tests on other substrates?

Answer: The 266 nm pump laser incident on our sample is partially reflected at the BTO surface (18%), it is mostly absorbed in the BTO thin film (70 %), and to a smaller extent it is absorbed in the SRO layer (12 %), as shown in Figure S13 and explained in Supplementary Note 7. As a result, GSO has no contribution to the optical response of the ferroelectric thin film because light is not expected to reach the substrate in a first place. In addition, even if our 266 nm pump laser, with photon energy 4.66 eV, reached the GSO substrate, it would be poorly absorbed, because GSO has a bandgap of $\sim 5.8 \text{ eV}$ [Derks] (see reference at the end of this Answer).

The choice of GSO as a substrate was dictated by the requirement of having a moderate in-plane compressive strain (-0.55 %) to the BTO thin film, resulting in the BTO film coherently strained to the substrate (Figure S16) with an out-of-plane tensile strain and monodomain. The requirement above aimed to avoid large epitaxial strain that could likely result in strain relaxation via dislocations and/or ferroelastic twinning. Such structural discontinuities would greatly complicate the analysis without satisfying the requirement above. Other possible substrates having a close lattice match with BTO are scandates anyway, e.g., DyScO_3 (DSO) or SmScO_3 (SSO). The latter have similarly large bandgap as GSO [Derks] ($E_g^{\text{DSO}} = 5.9 \text{ eV}$, $E_g^{\text{SSO}} = 5.6 \text{ eV}$), thus mostly transparent to our pump laser and have similar acoustic reflection coefficient ($R_z^{\text{DSO}} = 2 \%$, $R_z^{\text{SSO}} = 1 \%$), as compared to GSO ($R_z^{\text{GSO}} = 1 \%$, see Table S1). Therefore, choosing another scandate substrate would only change the strain imposed to BTO at equilibrium, with larger (DSO) or smaller (SSO) strain gradient along the out-of-plane direction [31], but would have negligible impact both on its optical response, and on the strain propagation.

To clarify the point raised by the Reviewer, in the Methods section (Time-resolved X-ray diffraction) at page 12, we replace the sentence below:

The transmittance profile of the 266 nm laser in our BTO/SRO/GSO sample is reported in Supplementary Note 6 A.

with the following sentence:

The 266 nm pump laser incident on our sample is partially reflected at the BTO surface (18%), it is mostly absorbed in the BTO thin film (70%), and to a smaller extent it is absorbed in the SRO layer (12%), as shown in Figure S13 and explained in Supplementary Note 7. As a result, GSO has no contribution to the optical response of the ferroelectric thin film.

Reference:

[Derks]: Derks et al., Band-gap variation in RScO₃ (R=Pr, Nd, Sm, Eu, Gd, Tb, and Dy): X-ray absorption and O K -edge x-ray emission spectroscopies, Phys. Rev. B 86, 155124, 2012 (DOI: [10.1103/PhysRevB.86.155124](https://doi.org/10.1103/PhysRevB.86.155124))

Comment: 4. The author mentioned DFT calculations several times in the paper (page 5, line 140 and page 7, lines 168 and 170), which involve the key Ti-O bond. Can some calculations be added to the paper to compare and verify the experimental results? This will significantly improve the quality of the article.

Answer: The cited time-dependent DFT calculations [32, 33] show independently and conclusively that photoexcited electrons weaken the bonds between Ti and apical O, more than the bonds between Ti and basal O. This is exactly what we see in our tr-SHG measurements, which provide the time evolution of the tensor elements referring to in-plane and out-of-plane induced electric dipole (Figure 2d). At the same time, the cited static DFT calculations [52, 53] show that photoexcited carriers screen long-range dipole-dipole interaction, thereby weakening the Ti-O bond responsible for the polar order. This is exactly what we see with the ultrafast decrease in polarization. Moreover, the cited works represent the state of the art in DFT calculations, they refer to the very same system studied here, namely BaTiO₃, and investigate the same range of photoexcited carrier density of our work, i.e., up to 0.2 e/f.u. (electron per formula unit, see Figure S12). Therefore, repeating these calculations ourselves would not add anything of value to the discussion.

In the Discussion section at page 8 (first paragraph), we add the following sentence to highlight the good agreement between our experiments and the cited references.

These theoretical studies [52, 53] investigated electron doping concentration per formula unit $n_{e/f.u.} \lesssim 0.2$, which corresponds precisely to the $n_{e/f.u.}$ range reached in our experiments (Figure S12).

Comment: 5. The paper points out that photogenerated electrons complete the initial polarization change within 350 fs, while the time scale of polarization recovery is significantly longer than the recovery time of reflectivity change. The authors do not seem to discuss the specific contributions of different scattering mechanisms in polarization recovery, and additional discussion is recommended.

Answer: In the section ‘Photoinduced ferroelectric polarization and electron dynamics’ (fourth paragraph, pages 6-7) we discuss the scattering mechanisms of electrons and phonons, as reported below. Since the polarization changes are mostly due to the dynamics of photoexcited carriers, the scattering mechanisms described below apply also to the polarization.

In both tr-SHG and tr-refl data, the time needed to reach the maximum relative change (350 fs) might be due to the thermalization of photoexcited electrons via electron-electron scattering. Subsequently, thermalized electrons, which are higher in the conduction band, move to the bottom of the conduction band, transferring energy to the phonon system, and recombining with holes in the valence band via electron-phonon scattering [4, 40, 45] or radiatively [46]. These processes are characterized by the recovery times τ_1 and τ_2 . Both τ_1 and τ_2 recovery constants of $\Delta I^{\text{P}_{\text{SHG}}}/I^{\text{P}_{\text{SHG},0}}$ are larger than those of $\Delta R/R_0$ because the dynamics of the spontaneous polarization (seen by tr-SHG) results from the convolution of the faster dynamics of photoexcited carriers (seen by tr-refl) and the slower dynamics of atoms.

The dynamics of photoexcited carriers (seen by tr-refl) and polarization (seen by tr-SHG) is mostly different after few tens of picoseconds. At this time scale, electrons have thermalized and mostly equilibrated with the lattice, except for some residual recombination (Figure 3h), thus there is no additional electron scattering mechanism responsible for the polarization recovery. In fact, the longer recovery time of the polarization is mostly due to the additional contribution of the structural dynamics, besides the electron dynamics.

We followed the recommendation of the Reviewer and made the corresponding discussion more explicit at page 7 (second last paragraph), as reported below:

After a few tens of picoseconds, the longer recovery time of the polarization, as compared to the photoexcited carriers, is due to the contribution of the much slower structural dynamics, affecting Δz_i , characterized by a recovery time beyond the 40 ps timescale, as seen in Figure 1d.

Comment: 6. *The I-V curves of ferroelectrics are important to discuss the photogenerated electrons. Thus, we suggest to add this results in the revised manuscript. Some literatures have discussed this issue, such as*

-Journal of Applied Physics 132 (2022) 224106.

-Journal of the European Ceramic Society 44 (2024) 5752–5764.

Answer: I-V curves are indeed a standard way to characterize the photocurrent under illumination in steady-state conditions. Typically, these measurements take place in the scale of seconds (see Figure 7d in Ref. 48). Measuring I-V curves on our sample implies adding a top electrode, which would modify the photocurrent, the polarization and the strain. Therefore, the corresponding I-V curves would not be comparable to the other results presented in our manuscript, simply because they would be measured essentially on a different sample.

While I-V curves are the standard tool to measure photocurrents in steady-state conditions, the typical tool to probe the ultrafast photogenerated carrier dynamics is by means of optical probe techniques, see e.g. the review article of Ref. 29. Therefore, we measured time-resolved reflectivity to probe the photoexcited carrier dynamics, as it is typically done in literature, see e.g. Refs. 38-40.

At the end of the section ‘Photoinduced ferroelectric polarization and electron dynamics’ (page 7), we add the paragraph below, which includes the references suggested by the Reviewer, and an extensive discussion of the points above.

Furthermore, we note that measuring current-voltage (I-V) curves is an alternative tool to measure photocurrents under illumination [48, 49]. These measurements are typically performed under steady-state conditions and require a top electrode. Given the absence of top electrode in our sample, adding it would certainly change the interfacial properties of our sample and influence the measured photogenerated electrons. As an example, it has been shown that the photocurrent may vary by more than 2 orders of magnitude, depending on the top electrode employed [50]. Furthermore, modifying the top interface would also modify the BTO ferroelectric polarization, at least locally near the surface, and possibly also deeper in the BTO layer, as investigated experimentally [11, 31, 51] and theoretically [47]. Finally, adding a top electrode would also affect the strain propagation because of the different acoustic reflection coefficient at the BTO surface (Table S1). Therefore, we probe the photoexcited carrier dynamics in our sample by tr-refl, as previously done in other works [38–40], and as typically done in ultrafast measurements [29].

Comment: 7. The authors mentioned that the Schottky interface effect can be ignored in this work. It is better to provide additional calculation or experimental data to support this conclusion. Provide additional analysis or literature support to enhance the persuasiveness.

Answer: Following the suggestion of the Reviewer, in the revised manuscript we provide additional analysis and literature support to motivate why the Schottky interface effect plays a minor role in this work. To address the comment of the Reviewer, we modify the third paragraph of the section 'Discussion' (page 8-9) and add a new section in the Supplementary Information, as reported below:

Second, ferroelectric thin films grown on a metal substrate form a Schottky interface, where the electron-hole pairs generated by light absorption are separated by the built-in voltage of the Schottky barrier. The photoexcited carriers are most efficiently separated in the depletion region of width w , where the built-in field exists. At the BTO/SRO interface [61] the width of the Schottky interface is $w \approx 5\text{nm}$, similar to the one of other ferroelectrics, e.g., $\text{Pb}(\text{Zr}_{0.2}\text{Ti}_{0.8})\text{O}_3/\text{SRO}$ interface [11, 62]. Outside the depletion region, carrier recombination dominates because of the small diffusion length $L_d \approx 2\text{nm}$ (Supplementary Note 6). Given the transmittance profile of the pump laser beam in our 34.5 nm thick BTO film (Supplementary Note 7 and Figure S13), only $\approx 5\%$ of the pulse energy deposited in BTO is absorbed in the depletion region of the Schottky barrier and may contribute to the variation of the polarization [11]. Therefore, in our case the Schottky interface effect is expected to play a minor role.

The new section in the Supplementary Information is reported below:

Supplementary Note 6. Estimation of the carrier diffusion length L_d

The diffusion length L_d , i.e., the distance that a charge carrier can move after generation and before it recombines, is determined by its mobility μ (Supplementary Note 1) and lifetime τ as $L_d = \sqrt{k_B T \mu \tau / e}$, where k_B , T and e are the Boltzmann constant, the temperature and the elementary charge, respectively. The electron mobility in BTO [9] is $\mu = 0.1 \text{ cm}^2\text{V}^{-1}\text{s}^{-1}$. The lifetime $\tau = 13 \text{ ps}$ is defined as the time required by $\Delta R/R_0$ to drop to $1/e$ of its maximum [31] (see Figure 2f). The temperature range near the BTO/SRO interface goes from 300 K to 417 K (Figure S3), depending on the pump-probe delay t . The parameters above yield L_d in the range 1.8 – 2.2 nm, with a maximum of 3nm near the surface, where T reaches the maximum temperature of 847 K.

Comment: 8. The author cited and compared many previous studies in analyzing the mechanism, but has not yet clearly explained the key differences between this study and existing reported. It is recommended to clarify the highlights of the article in the revised manuscript.

Answer: There have been indeed several studies investigating how strain and polarization change with time upon photoexcitation of ferroelectric materials. However, none of these studies has ever looked at both polarization and strain, instead, they measured either one or the other, and assumed they go hand in hand. In addition, previous reports have employed significantly lower peak power intensity of the pump laser, which prevents a direct manipulation of the polarization.

We follow the suggestion of the Reviewer and highlight the key differences between this study and the previous reports in the Discussion at page 10, as reported below:

In summary, we demonstrate that, with an above-bandgap laser excitation, an ultrafast decoupling of spontaneous polarization and strain can be achieved and measured. The key difference to previous studies is that we measure both the polarization and the strain, and thus we are able to observe the ultrafast decoupling of polarization and strain taking place already within 350 fs and lasting at least for several tens of picoseconds. Another important difference to previous reports is that we employ a larger peak power intensity of the pump laser (39 GWcm^{-2} , see Supplementary Note 4) with excitation energy above the bandgap. This is important to excite a sufficient number of electrons from the O 2p-derived valence band to the Ti 3d-derived conduction band, reaching $\approx 0.2 \text{ e/f.u.}$ (Figure S12), and to induce the change in polarization, as predicted by theory [52, 53].

We highlight the importance of the high peak power which excites a sufficient number of electrons from the O 2p-derived valence band to the Ti 3d-derived conduction band also in the abstract as reported below:

Here, upon an above-bandgap laser excitation of the prototypical ferroelectric BaTiO₃, with sufficiently high peak power intensity of a few tens of GW cm^{-2} , we induce and measure an ultrafast decoupling between polarization and strain that begins within 350 fs, by softening Ti-O bonds via charge transfer, and lasts for several tens of picoseconds.

and in the second paragraph of the introduction:

To explore this scenario, we probe the out-of-plane strain and the spontaneous polarization of the prototypical ferroelectric BaTiO₃ upon above-bandgap absorption of ultrashort UV light pulses with peak power intensity of a few tens of GW cm^{-2} (Figure 1a).

Reviewer #2 (Remarks to the Author):

This study investigates the ultrafast decoupling of polarization and strain in the ferroelectric material BaTiO₃ (BTO) under above-bandgap UV light excitation. Using time-resolved X-ray diffraction (tr-XRD) and second-harmonic generation (tr-SHG), the researchers examine the separate dynamics of the lattice (out-of-plane strain), spontaneous polarization, and photoexcited carriers. They show that the decoupling occurs within 350 fs after excitation, attributed primarily to the effect of photoexcited electrons that influence the polarization, while the strain response is slower, dictated by the lattice dynamics. The results challenge the assumption that polarization and strain are always coupled in ferroelectrics, especially out of equilibrium, and demonstrate the potential for using light to manipulate the polarization independently of strain.

Comment: 1. *The authors should supplement their analysis with SAED or STEM data to directly confirm the single-domain nature of the BTO films. This would strengthen the argument and address potential ambiguities in the current dataset.*

Answer: The single-domain nature of the BTO film, with polarization pointing towards the surface, was determined by piezoresponse force microscopy (PFM), as shown in Figure S18. The latter is a non-destructive technique that can probe areas of several microns squared and is sensitive to the average polarization of the entire film, for thicknesses of a few tens of nanometers as in our case (see [Harnagea, Eng], references are at the end of the this Answer). In addition, reciprocal space mapping (RSM) data (Figure S16) demonstrate that there is no secondary phase or twinning structure, indicating that the BTO is fully constrained to the underlying SRO film and GSO substrate. We note that RSM probes an area of 1.3 mm x 0.6 mm. Furthermore, PFM and RSM are the standard technique employed to determine the single-domain nature of ferroelectrics, as an example we refer to the two articles [27, 28] suggested by Reviewer 3. In addition, we also performed SHG measurements at different azimuthal angles (Figure S19). These data support the out-of-plane nature of the spontaneous polarization and the absence of a net in-plane polarization component. With SHG a large area of few thousands of microns squared is probed.

In summary, all the characterization techniques typically used to determine the domain nature of a ferroelectric thin film have been employed and unambiguously converge to the single-domain nature of our BTO film. We note that PFM, RSM and SHG are all non-destructive and probe large areas of the sample.

In addition, we follow the suggestion of the Reviewer and perform STEM measurements (Supplementary Note 9 and Figure S17) to supplement the analysis presented above. Also STEM confirms the single crystalline nature of the BTO thin film, as shown in Figure S17a.

We report below the additional Supplementary Note 9, where STEM data are presented, followed by Figure S17:

Supplementary Note 9. Scanning transmission electron microscopy

High resolution scanning transmission electron microscopy (HR-STEM) was performed on a Cs probe corrected FEI Titan3 G2 60–300 microscope (Thermo Fisher Scientific), equipped with a GIF Quantum ERS from Gatan for electron energy loss spectroscopy (EELS), and a window-less, energy-dispersive X-ray (EDX) system (Super-X) using a four-quadrant silicon drift detector (SDD). The microscope was operated at 300 kV in scanning mode. The electron beam current was set to approximately 50 pA and a convergence semi-angle of 15 mrad was used. The angular range for high-angle annular dark-field (HAADF) imaging was 62.2 to 214.0 mrad. All data for imaging were taken from an area with a relative thickness of 0.6 – 0.7 in units of inelastic mean free path measured by EELS. Image acquisition and analysis was performed using GMS 3 (version 3.60) by Gatan. EDX spectra were acquired in a thicker area of 0.8 – 1.0 in units of inelastic mean free path using Velox inherent drift correction, and EDX data were analyzed using Velox by ThermoFisher (version 3.5).

The preparation of the lamella for STEM measurements was done by conventional Focused Ion Beam (FIB) milling on a FEI Nova 200 FIB-SEM, using 10 kV acceleration voltage with a Ga⁺ ion beam current of 50 pA in order to minimize sample damage.

Figure S17a shows the BTO/SRO interface with single crystalline BTO and no visible stress or dislocations. The BTO/SRO interface is not atomically sharp, but steps do not exceed one unit cell. The SRO/GSO interface is flat and atomically sharp (Figure S17b). The dark area along the SRO/GSO interface most likely corresponds to differences in surface amorphization arising during the preparation of the lamella for STEM, as elemental analysis did not reveal any compositional variations in this region. At the BTO/SRO interface (Figure S17c), EDX elemental maps of Ba (Figure S17d), Ti (Figure S17e), Sr (Figure S17f) and Ru (Figure S17g) confirm that interdiffusion across the interface is below the detection limit.

Figure S17 STEM images and EDX elemental maps. HR-STEM HAADF image of BTO/SRO interface (a), and overview image showing both BTO/SRO and SRO/GSO interfaces (b). HR-STEM HAADF image of the BTO/SRO interface (c) and the corresponding EDX elemental maps of Ba L lines (d), Ti K lines (e), Sr K lines (f), and Ru K lines (g). All acquired images are unfiltered. EDX spectra were acquired with a dwell time of 25 μ s and an acquisition time of 580 s.

We highlight the single-domain nature of the BTO thin film in the section ‘Sample preparation and characterization’ in Methods, as reported below:

We determine the out-of-plane lattice parameters $c_{\text{BTO}} = 4.074 \text{ \AA}$, $c_{\text{SRO}} = 3.934 \text{ \AA}$, and $c_{\text{GSO}} = 3.964 \text{ \AA}$ by means of the reciprocal space map (RSM) shown in Figure S16. The in-plane lattice parameter $a = 3.970 \text{ \AA}$, common to BTO, SRO and GSO, indicates that both BTO and SRO thin films are coherently strained to the substrate. The in-plane compressive strain of -0.55% imposed to BTO by the substrate is calculated by comparing in-plane lattice parameter a with the respective bulk value $a_{\text{b,BTO}} = 3.992 \text{ \AA}$ [67], according to $(a - a_{\text{b,BTO}})/a_{\text{b,BTO}}$. The absence of satellite peaks in RSM data (Figure S16) suggests the existence of a BTO monodomain. The single-crystalline nature of the BTO film with no visible stress or dislocations is demonstrated by scanning transmission electron microscopy (STEM) data (Figure S17a and Supplementary Note 9). Both SRO/GSO and BTO/SRO interfaces are of high-quality. The former is atomically sharp (Figure S17b), and the latter has steps not exceeding one unit cell with interdiffusion below the detection limit (Figure S17c-g). The single-domain nature of the BTO film is further demonstrated by means of piezoresponse force microscopy (PFM, Figure S18), which provides the spontaneous polarization of the as-grown sample P_s pointing upward (toward the surface). Furthermore, the independence of both SHG polar plots of $I^{\text{P}}_{\text{SHG}}(\phi)$ and $I^{\text{S}}_{\text{SHG}}(\phi)$ from the azimuthal angle γ (Figure 2a) confirms the out-of-plane nature of the spontaneous polarization of our BTO sample, and the absence of a net in-plane component of the polarization (Figure S19).

References:

[Harnagea]: Harnagea, C. & Pignolet, A. Challenges in the Analysis of the Local Piezoelectric Response. In Alexe, M. & Gruverman, A. (eds.) Nanoscale Characterisation of Ferroelectric Materials: Scanning Probe Microscopy Approach, NanoScience and Technology, 45–85 (Springer, 2004) DOI: [10.1007/978-3-662-08901-9_2](https://doi.org/10.1007/978-3-662-08901-9_2)

[Eng]: Eng, L. M. et al. Local dielectric and polarization properties of inner and outer interfaces in PZT thin films. Integr. Ferroelectr. **62**, 13–21 (2004) DOI: [10.1080/10584580490460277](https://doi.org/10.1080/10584580490460277)

Comment: 2. The manuscript describes the use of In Situ X-ray Diffraction (IXRD) and optical methods to analyze photoinduced lattice strain. However, no Transmission Electron Microscopy (TEM) data is provided to directly observe atomic displacements, such as changes in Ti-O bond lengths. The authors should include TEM measurements to directly observe Ti-O bond length changes and validate the XRD-derived conclusions about photoinduced lattice strain.

Answer: We thank the Reviewer for the interesting suggestion and we structure our reply in three parts addressing (i) the application of TEM to our sample in relation to existing tr-XRD, tr-SHG and tr-refl data, (ii) the determination of Ti-O bond length changes and (iii) the photoinduced strain.

(i)-a. First, the proposal of the Reviewer to perform TEM to directly observe Ti-O bond length changes is certainly very interesting, however the results obtained from such measurements on our sample could not be compared to the existing tr-XRD, tr-SHG and tr-refl data for the following reasons. Preparing a lamella for TEM analysis of a large-bandgap material such as BTO using a focused ion beam (FIB) requires the deposition of a thin conductive carbon layer over the entire surface of the ferroelectric thin film to mitigate charging effects. This is followed by the local deposition of a Pt/C layer in the FIB to protect the sample from the Ga ion beam. Changing the top interface of our sample will certainly also modify the BTO ferroelectric polarization, at least locally near the surface, and possibly also deeper in the BTO layer, as investigated experimentally [11, 31, 51] and theoretically [47]. This has a major influence on the Ti-O bond lengths. As a result, a hypothetical TEM experiment would essentially probe the Ti-O bond length changes of another sample (metallic-layer/BTO/SRO/GSO), and the corresponding results could not be compared to the present tr-XRD, tr-SHG and tr-refl data performed on BTO/SRO/GSO without any metallic layer on top.

(i)-b. Furthermore, the boundary conditions for the strain development in a lamella are significantly different as compared to the sample investigated in this work both along the (I) out-of-plane and (II) in-plane directions. (I) Along the out-of-plane direction, our sample BTO/SRO/GSO has an acoustic reflection coefficient $R_z = -1$ at the BTO-vacuum interface (Table S1), thus the strain wave is fully reflected at the BTO-vacuum interface. In contrast, the presence of a metallic layer with $R_z \neq -1$, would significantly alter the propagation of the photoinduced strain generated in BTO. In fact, part of the strain wave travelling in BTO would be transmitted through the metallic layer and an additional strain wave would originate the surface of the metallic layer. (II) Along the in-plane direction, the thickness of the lamella ~ 50 nm introduces new boundaries, which alter the strain propagation as compared to the thin film sample. As a consequence of points (I) and (II), the results of a hypothetical TEM experiment could not be directly used to validate the conclusions about the photoinduced lattice strain obtained by tr-XRD.

(i)-c. Moreover, tr-XRD is a non-destructive technique probing a relatively large area, namely in our case $140 \mu\text{m} \times 100 \mu\text{m}$ (see Methods), thus several orders of magnitude larger compared to the area probed by TEM, typically $10\text{-}15 \mu\text{m} \times 50\text{-}100 \text{nm}$. Therefore, tr-XRD is representative of a much larger sample volume, which is also similar to the one probed by tr-SHG and tr-refl (see

Methods). This makes the results obtained by the three different measurements (tr-XRD, tr-SHG, tr-refl) more comparable.

(i)-d. Finally, we note that, while TEM is certainly a powerful tool to investigate static and photoinduced lattice strain, so far, the great majority of time-resolved studies employ tr-XRD, see Refs. [5-6, 8, 12-18] for examples relating ferroelectrics and multiferroics, and none of the works above included TEM measurements to validate the results obtained from tr-XRD.

(ii) Second, we turn now to discuss the Ti-O bond length changes questioned by the Reviewer. After the absorption of the pump laser we observe a gradual intensity decrease near the BTO (001) diffraction peak center (Figure 1c), reaching a minimum after ~ 3.5 ps. The drop in diffraction intensity goes hand in hand with the initial contraction of the lattice. When the lattice contracts, one should expect a reduction of the displacement between Ti and the center of the O octahedron. This is exactly what the diffraction intensity drop indicates. In fact, simulations based on dynamical theory of diffraction reproduce quite accurately the diffraction intensity drop as a result of 8 pm change in the Ti-O displacement $\Delta_{\text{Ti-O}}$. We note that the simulations employed here, have been extensively and successfully employed to model the BTO (001) diffraction peak of similar BTO thin films, grown on different scandate substrates (SmScO_3 , GdScO_3 , DyScO_3) with a SRO layer in between [31]. The reference where the simulations are presented in great detail is now added to the revised manuscript:

Specifically, simulations based on the dynamical theory of diffraction [31] (Figure S2) exclude the Debye-Waller effect and show that a decrease in the displacement $\Delta_{\text{Ti-O}}$ between the Ti atom and the center of the O octahedron by 8 pm can model the measured maximum change in peak diffraction intensity.

The following text has been added to the caption of Figure S2:

Details of the simulations based on the dynamical theory of diffraction can be found in Ref. [1].

(iii) Third, we move to discuss the photoinduced lattice strain questioned by the Reviewer. The shift of the BTO (001) diffraction peak (Figure 1b) shows unambiguously the contraction/expansion of the lattice along the out-of-plane direction (Figure 1d). These experimental data alone would be already sufficient to draw the main conclusion of the paper, which is the ultrafast decoupling between polarization and strain. However, we go one step further and model the evolution of the strain upon photoexcitation. The model is summarized in the last paragraph of the section 'Photoinduced structural dynamics' at page. 4-5, and further details are reported in Supplementary Notes 1, 2, and 3. In short, we consider the pump pulse energy absorbed by our sample, calculate the electron and phonon temperature, and subsequently solve the strain wave equation. We obtain an excellent fit of the experimental strain data (Figure 1d and Figure S4) with only 3 fit parameters, whose values are consistent with expectations and literature (Supplementary Note 2). Additionally, based on the resulting strain

profiles at different pump-probe delays t , we calculate the corresponding diffraction curves and find a very good agreement with the experimental ones, as shown in Figure S7 e-f, and Figure S8 e-f, for the pump fluences 2.7 mJcm^{-2} and 1.4 mJcm^{-2} , respectively. The agreement between calculated and experimental diffraction curves at different pump-probe delays, together with the excellent fit of the time-dependent strain data, validate the model employed to describe the photoinduced strain data measured by tr-XRD.

In the revised manuscript, at the end of the section 'Photoinduced structural dynamics' in Results, we add the following text:

The validity of the model employed to fit the strain $\eta(z,t)$ data is further corroborated by the good agreement between the experimental BTO (001) diffraction peaks measured at different pump-probe delays t and the corresponding calculated diffraction curves, based on the strain profiles as a function of delay t and distance z from the BTO surface (Figure S7 and Figure S8).

Comment: 3. The manuscript does not address the potential influence of the interface structure between BTO/SRO/GSO heterostructures on the transfer of photoinduced strain. Features such as dislocations or stress concentrations at the interfaces could significantly affect how strain propagates through the system. Understanding these interface characteristics is critical for interpreting the observed photoinduced strain behavior.

If these aspects are adequately addressed, the paper could indeed be suitable for publication in Nature Communications.

Answer: We thank the reviewer for considering our work suitable for publication in Nature Communication, after addressing the aspects above. To answer the comment of the Reviewer, we performed STEM measurements on our sample. The corresponding data (Supplementary Note 9 and Figure S17a-b) show the high quality of both BTO/SRO and SRO/GSO interfaces with no visible dislocations or stress concentrations. Specifically, the BTO/SRO interface has steps not exceeding one unit cell, while the SRO/GSO interface is flat and atomically sharp. Furthermore, EDX elemental maps of Ba, Ti, Sr and Ru show that the interdiffusion across the BTO/SRO interface is below the detection limit (Figure S17c-g). The high quality of BTO/SRO and SRO/GSO interfaces, without dislocations or stress concentrations, justifies the employment of the acoustic reflection coefficient reported in Table S1, which are based on mass density and longitudinal sound velocity available in literature. Finally, the good fit of the time evolution of the average strain (Figure 1d and Figure S4), and the good agreement between calculated and experimental diffraction curves (Figure S7e-f and Figure S8e-f), validate the strain propagation model employed here.

Reviewer #3 (Remarks to the Author):

This paper investigates the ultrafast decoupling of polarization and strain in ferroelectric BaTiO₃ upon above-bandgap laser excitation. Through a combination of tr-XRD, tr-SHG, and tr-refl, the authors observe and measure the dynamics of lattice parameters, spontaneous polarization, and photoexcited carrier density. The study reveals that photoexcited electrons play a dominant role in determining the spontaneous polarization out of equilibrium, leading to an ultrafast decoupling between polarization and strain. The paper presents novel findings and employs advanced experimental techniques, making it suitable for publication in Nature Communications, provided the following issues are addressed:

Comment: 1. The format of the formula in line 152 is misleading. The authors wrote it as $P_s(t) = 1/V \sum_i q_i(t) \Delta z_i(t)$, as the summation appears to be in the denominator. This should be clarified to ensure the summation is correctly placed in the numerator. Additionally, the interpretation of the SHG intensity drop and reflectivity increase based on this formula may need reconsideration.

Answer: We thank the Reviewer for the generally positive evaluation of our work. In the revised manuscript at page 7, we add the brackets around the $1/V$ term to avoid misinterpretation of the formula, as reported below:

To interpret the SHG intensity drop and the reflectivity increase, it is useful to express the spontaneous polarization as [22, 47]: $P_s(t) = (1/V) \sum_i q_i(t) \Delta z_i(t)$, where V is the volume of the unit cell, $q_i(t)$ is the **local** charge and $\Delta z_i(t)$ is the out-of-plane displacement of atom i **from the high symmetry position**.

The interpretation of the SHG intensity drop based on the formula above follows from the proportionality $I_{\text{SHG}} \propto |\chi^{(2)}_{ijk}|^2 \propto |P_s|^2$, which is well established and clearly stated in the seminal paper of Fiebig et al. [44]. This proportionality was later used in many other publications, see for example [11, 21, Nova, see reference at the end of this Answer]. The reference of the review paper of Fiebig et al. [44] is now present in the revised manuscript:

The proportionality $I_{\text{SHG}} \propto |\chi^{(2)}_{ijk}|^2 \propto |P_s|^2$ gives direct access to the magnitude of the spontaneous polarization [44].

In short, changes in the SHG intensity reflect changes in the local charges (q_i) and the atomic structural dynamics (Δz_i). On the other hand, changes in reflectivity relate to changes in the photoexcited carrier density n_e , as it is commonly done in literature, see for example Ref. [38-40] and the review article [29].

Reference:

[Nova]: Nova, T.F. et al., Metastable ferroelectricity in optically strained SrTiO₃, Science 364, 1075 (2019) DOI: [10.1126/science.aaw4911](https://doi.org/10.1126/science.aaw4911)

Comment: 2. The authors attempt to directly correlate the reduction in SHG intensity with the decrease of ferroelectric polarization. However, while the measured SHG intensity is proportional to the square of $\chi_{ijk}^{(2)}$, its magnitude is also influenced by the material's permittivity. The observed changes in tr-refl also indicate alterations in the dielectric characteristics of BTO. How can the authors exclude the relationship between the reduction in SHG signal intensity and the changes in dielectric properties? This point should be addressed in the discussion.

Answer: The correlation between SHG intensity and ferroelectric polarization follows from the well-known proportionality $I_{\text{SHG}} \propto |\chi_{ijk}^{(2)}|^2 \propto |P_s|^2$, discussed in the Answer to Comment 1. We agree with the Reviewer that $\chi^{(2)}$ depends also on the material permittivity ϵ , which is related to the electric susceptibility χ_e , as detailed in the revised manuscript and reported below. In fact, we do not exclude a relationship between the reduction in the SHG signal intensity and changes in dielectric properties. In contrast, our data indicate that changes in $\Delta I_{\text{SHG}}^{\text{P}}/I_{\text{SHG},0}^{\text{P}}$ (Figure 2e) are primarily due to changes in the dielectric properties, which manifest in the time dependence of $\Delta R/R_0$ (Figure 2f), and secondarily are due to the structural dynamics seen by tr-XRD (Figure 1d). In our experiments, changes in the dielectric properties of BTO result from the photoexcitation of electrons from the O 2p-derived valence band to the Ti 3d-derived conduction band, which leads to an increase in the photoexcited carrier density n_e . In the manuscript, we discuss $\Delta I_{\text{SHG}}^{\text{P}}/I_{\text{SHG},0}^{\text{P}}$ and $\Delta R/R_0$ in terms of changes in polarization P_s and photoexcited carrier density n_e , as it is typically done in literature [29, 38-40].

We follow the suggestion of the Reviewer, and at the last paragraph of the section 'Photoinduced ferroelectric polarization and electron dynamics' (Results) we add the sentence:

An equivalent interpretation of SHG intensity and reflectivity in terms of nonlinear SHG coefficient and electric susceptibility is reported in Methods.

In the section Methods of the revised manuscript we discuss the relationship between SHG intensity, reflectivity and dielectric properties, as reported below:

Interpretation of SHG intensity and reflectivity

In the section Results we discuss SHG intensity and reflectivity as proportional to the spontaneous polarization P_s and the photoexcited carrier density n_e , respectively. In the following, we show an equivalent interpretation of SHG intensity and reflectivity in terms of nonlinear SHG coefficient and electric susceptibility. In general, the material permittivity ϵ relates the applied electric field \mathbf{E} to the electric displacement field \mathbf{D} according to $\mathbf{D} = \epsilon\mathbf{E} = \epsilon_0\mathbf{E} + \mathbf{P}$, where ϵ_0 is the vacuum permittivity and \mathbf{P} is the induced electrical polarization. Following the notation in Ref.42, \mathbf{P} can be expressed as:

$$\mathbf{P}(t) = \mathbf{P}_0 + \epsilon_0\chi_e\mathbf{E}(t) + \chi^{(2)}\mathbf{E}(t)^2, \quad (3)$$

where \mathbf{P}_0 is a time-independent constant polarization. The second term in equation (3) depends linearly on $\mathbf{E}(t)$ and induces electric dipole oscillations at ω , i.e., the same oscillation frequency as the incident electric field of light, thus it is responsible for the optical reflectivity. In equation (3), $\chi_e = n^2_{\omega} - 1$ is the electric susceptibility of the material and n_{ω} is the refractive index at ω . The third term in equation (3) has a quadratic dependence on $\mathbf{E}(t)$ and induces electric dipole oscillation at 2ω , twice the frequency of the incident light, thus it is responsible for the second harmonic generation process. In particular, if we ignore the tensor form, the nonlinear SHG coefficient can be expressed as [42]:

$$\chi^{(2)} \propto A\chi_e^2(\omega)\chi_e(2\omega), \quad (4)$$

where $\chi_e(2\omega) = n^2_{2\omega} - 1$, with $n_{2\omega}$ being the refractive index at 2ω , and A is a structural parameter related to the atomic displacements Δz_i from the high symmetry positions. In materials with a center of inversion symmetry, $A = 0$, $\chi^{(2)} = 0$, and the second order term of the induced polarization vanishes. Therefore, tr-SHG measurements are sensitive to both electronic (via $\chi_e(\omega)$ and $\chi_e(2\omega)$) and structural (via A) changes in the material. At the same time, tr-refl measures changes in the electric susceptibility $\chi_e(\omega)$, which, in our work, result from the photoexcitation of electrons from the O 2p-derived valence band to the Ti 3d-derived conduction band, while structural changes are monitored by tr-XRD. The SHG intensity is proportional to $|\chi^{(2)}|^2$, and thus to a combination of both electronic and structural contributions. The time-evolution of $\Delta I^{P_{\text{SHG}}}/I^{P_{\text{SHG},0}}$ and $\Delta R/R_0$ (Figure 2e-f) indicate that $\chi^{(2)}$ is mostly determined by the changes in electric susceptibility $\chi_e(\omega)$, and to a smaller extent to the structural changes in A . This is equivalent to say that changes in the spontaneous polarization P_s are mainly determined by the photoexcited electrons and in minor part to the structural dynamics of the lattice, as discussed in the section Photoinduced ferroelectric polarization and electron dynamics (Results).

Comment: 3. Please ensure that the figures and sections in the Supplementary Material are arranged in the order they are referenced in the main text. This will improve the coherence and ease of reference for readers.

Answer: All the items (Figures, Tables, Supplementary Notes and Supplementary Figures) present in the Supplementary Information are now listed following the order of reference in the main text.

Comment: 4. The authors should clarify why charge transfer leads to a reduction in the Born effective charge. This would enhance understanding for a broader audience.

Answer: We follow the suggestion of the Reviewer and clarify why charge transfer leads to a reduction in the local charge in the second last paragraph of the revised manuscript at page 7:

The above-bandgap photoexcitation transfers electrons from the O 2p-derived valence band to the Ti 3d-derived conduction band of BTO. This charge transfer from O to Ti atoms reduces both the local negative charge at the O site (q_O) and the local positive charge at the Ti site (q_{Ti}). We attribute the increase in R to the increase in photoexcited carrier density n_e [38–40], which decreases q_i , while P_s results from changes in both q_i and Δz_i . This offers the opportunity to manipulate P_s by modifying q_i , independently from the atomic displacements Δz_i .

Comment: 5. In line 203, the phrase “...to the top of the conduction band...” should be “... to the bottom of the conduction band...”

Answer: The sentence has been corrected in the revised manuscript, see third paragraph at page 9 reported below:

At the same time, the relaxation of photocarriers to the bottom of the conduction band and their recombination ...

Comment: 6. References related to strain manipulating ferroelectric polarization (DOI: 10.1002/advs.202417165 and DOI: 10.1002/advs.202307571) should be included to enrich the introduction part.

Answer: We add the references above suggested by the Reviewer in the introduction, as shown below:

Decoupling the polarization from the strain would remove the constraint of sample design [24, 26] or strain tuning [27, 28] to achieve specific properties, and, at the same time, would provide a more effective and ultrafast knob to manipulate the polarization by light.

We also add the same references in the last paragraph of the Discussion at page 10, as reported below:

The decoupling of polarization from the strain offers the opportunity to change paradigm from strain engineering [27, 28, 66] to light-induced polarization engineering, thereby lifting the constraint of selecting among a limited number of substrates [24] or designing freestanding membranes [26] to tune the spontaneous polarization.

Point-by point response to the reviewers

Reviewer #3 (Remarks to the Author):

The authors have addressed the issues raised by myself and the other reviewers. In its current form, I believe the manuscript now meets the publication standard of Nature Communications.

Comment: Only one minor note remains: '4mm' should be italicized.

Answer: We follow the remark of Reviewer 3 and italicize '4mm'.